



# Formation of condensable organic vapors from anthropogenic and
# biogenic VOCs is strongly perturbed by NOx in eastern China
Yuliang Liu[1,2], Wei Nie[1,2*], Yuanyuan Li[1,2], Dafeng Ge[1,2], Chong Liu[1,2], Zhengning
Xu[1,2,5], Liangduo Chen[1,2], Tianyi Wang[1,2,10], Lei Wang[1,2], Peng Sun[1,2], Ximeng Qi[1,2],
Jiaping Wang[1,2], Zheng Xu[1,2], Jian Yuan[1,2], Chao Yan[3], Yanjun Zhang[3,9], Dandan
Huang[4], Zhe Wang[6], Neil M. Donahue[7], Douglas Worsnop[8], Xuguang Chi[1,2], Mikael
Ehn[3], and Aijun Ding[1,2]
[1] Joint International Research Laboratory of Atmospheric and Earth System Sciences, School of
Atmospheric Sciences, Nanjing University, Nanjing, Jiangsu Province, China
[2] Collaborative Innovation Center of Climate Change, Jiangsu Province, China
[3] Institute for Atmospheric and Earth System Research/Physics, Faculty of Science, University of
Helsinki, Helsinki, Finland
[4] State Environmental Protection Key Laboratory of Formation and Prevention of Urban Air
Pollution Complex, Shanghai Academy of Environmental Sciences, Shanghai, China
[5] College of Environmental & Resource Sciences, Zhejiang University, Zhejiang, China
[6] Department of Civil and Environmental Engineering, the Hong Kong Polytechnic University,
Hong Kong SAR
[7] Center for Atmospheric Particle Studies, Carnegie Mellon University, Pittsburgh, PA, USA
[8] Center for Aerosol and Cloud Chemistry, Aerodyne Research Inc., Billerica, MA, USA
[9] Univ. Lyon, Université Claude Bernard Lyon 1, CNRS, IRCELYON, 69626, Villeurbanne, France
[10] Meteorological Service Center of Hubei Province, Wuhan, 430074
## Abstract
Oxygenated organic molecules (OOMs) are the crucial intermediates linking volatile
organic compounds (VOCs) to secondary organic aerosol (SOA) in the atmosphere, but
understandings on the characteristics of OOMs and their formations from VOCs are
very limited. Ambient observations of OOMs using recently developed mass
spectrometry techniques are still limited, especially in polluted urban atmosphere where
VOCs and oxidants are extremely variable and complex. Here, we investigate OOMs,
measured by a nitrate-ion-based chemical ionization mass spectrometer at Nanjing in
eastern China, through performing positive matrix factorization on binned mass spectra
(binPMF). The binPMF analysis reveals three factors about anthropogenic VOCs
(AVOCs) daytime chemistry, three isoprene-related factors, three factors about biogenic
VOCs (BVOCs) nighttime chemistry, and three factors about nitrated phenols. All
factors are influenced by NOx in different ways and to different extents. Over 1000 non-
nitro molecules have been identified and then reconstructed from the selected solution





of binPMF, and about 72% of the total signals are contributed by nitrogen-containing
OOMs, mostly regarded as organic nitrates formed through peroxy radicals terminated
by nitric oxide or nitrate-radical-initiated oxidations. Moreover, multi-nitrates account
for about 24% of the total signals, indicating the significant presence of multiple
generations, especially for isoprene (e.g., $C_5H_{10}O_8N_2$ and $C_5H_9O_{10}N_3$). Additionally, the
distribution of OOMs concentration on carbon number confirm their precursors driven
by AVOCs mixed with enhanced BVOCs during summer. Our results highlight the
decisive role of $NO_x$ on OOMs formation in densely populated areas, and encourage
more studies on the dramatic interactions between anthropogenic and biogenic
emissions.
**1 Introduction**
Secondary organic aerosol (SOA), as an important and complex component of
submicron particles (Zhang et al., 2007;Jimenez et al., 2009;Huang et al., 2014), is fully
involved in affecting climate (Intergovernmental Panel on Climate, 2014) and causing
health risks (Nel, 2005;Lim et al., 2012). Volatile organic compounds (VOCs) are
ubiquitous in the atmosphere and are recognized as main precursors of SOA (Hallquist
et al., 2009;Ziemann and Atkinson, 2012). However, the missing intermediate processes
from VOCs to SOA are yet to be elucidated (Hallquist et al., 2009;Ehn et al., 2014).
Benefitting from the state-of-the-art measurement technics (Bertram et al.,
2011;Jokinen et al., 2012;Lee et al., 2014), many previously unreported oxygenated
organic molecules (OOMs), as intermediates from VOCs to SOA (Ziemann and
Atkinson, 2012), have been discovered. Among OOMs, highly oxygenated organic
molecules (HOMs), first observed in the gas phase at a boreal forest site (Ehn et al.,
2010;Ehn et al., 2012) and have been reviewed by Bianchi et al. (2019), are so
functionalized and low volatile that they can participate at the beginning of new particle
formation (NPF) by stabilizing sulfuric acid (Kulmala et al., 2013;Riccobono et al.,
2014) or through clustering alone (Kirkby et al., 2016;Bianchi et al., 2016), and
condense on existing particles to be responsible for a large fraction of SOA (Ehn et al.,
2014). In addition to conventional VOCs-to-OOMs mechanisms summarized in the
Master Chemical Mechanism (MCM) (http://mcm.york.ac.uk/, last access: 09 February
2021), recent studies have proposed new pathways, such as autoxidation (Crounse et
al., 2013;Jokinen et al., 2014) and multigenerational oxidation (Rollins et al.,
2012;Wang et al., 2020b), to form condensable vapors by adding oxygen atoms
efficiently. The productions of OOMs, especially HOMs, from precursors such as
monoterpenes (Ehn et al., 2014;Jokinen et al., 2015;Kirkby et al., 2016;Berndt et al.,
2016), sesquiterpenes (Richters et al., 2016), isoprene (Jokinen et al., 2015;Zhao et al.,
2020) , aromatics (Wang et al., 2017;Molteni et al., 2018;Garmash et al., 2020), and
alkanes (Wang et al., 2021) have been investigated in laboratories by using the chemical
ionization atmospheric pressure interface time-of-flight mass spectrometer with nitrate
reagent ions (nitrate CI-APi-TOF).



New insights and a general understanding about OOMs have been attained, yet many
critical details about OOMs formation and properties need to be addressed. First, the
current kinetic descriptions of OOMs obtained from experiments are still limited, such
as the lack of individual H-shift rates for autoxidation and of reaction rates of multi-
generational products with oxidants. Furthermore, the complexity of the real
atmosphere makes it more difficult to apply experimental results to ambient
environments. The precursors compete for oxidants and vice versa, and their products
will interact mechanistically in mixtures of atmospheric vapors (McFiggans et al.,
2019;Heinritzi et al., 2020). However, in the laboratory we usually study simple
systems with a single precursor and a single oxidant. Moreover, most experiments are
carried out for environments dominated by biogenic VOCs (BVOCs), while
anthropogenic emissions receive less attention. In addition to classic anthropogenic
VOCs (AVOCs), large amounts of primary emissions of oxygenated VOCs are also
present in urban areas (Karl et al., 2018). The effect of NOx on OOMs is another key
issue. $NO_x$ can terminate peroxy radicals ($RO_2$), outcompeting autoxidation
propagation reactions and other bimolecular reactions ($RO_2 + RO_2$, $RO_2 + HO_2$), and
change the products distribution, and consequently, size-dependently modulate the
growth rates of organic aerosol particles (Yan et al., 2020). Additionally, $NO_x$
contributes non-linearly to atmospheric oxidants, which also influence the productions
of OOMs (Pye et al., 2019). It is anticipated that $NO_x$ plays a varied role in the
formations of OOMs as well as SOA in different environments.
Therefore, more extensive OOMs observations are needed to validate the atmospheric
implications of experiments, to couple with the global or regional model, and finally to
comprehensively understand the fate of OOMs in the atmosphere. Until now, only a
few ambient observations of OOMs using nitrate CI-APi-TOF have been reported
(Bianchi et al., 2019), and almost all of them focus on rural or forested or remote
atmospheres (Yan et al., 2016;Massoli et al., 2018;Zhang et al., 2020;Beck et al., 2021).
The Yangtze River delta (YRD) is one of the most developed regions in eastern China.
Fine particulate matter, with an aerodynamic diameter smaller than 2.5 μm ($PM_{2.5}$), has
been significantly reduced in eastern China after the implementation of "Action Plan
for Air Pollution Prevention and Control" since 2013 (Ding et al., 2019), while
(secondary) organic aerosol are still much more abundant than in clean areas (Zhang et
al., 2017;Sun et al., 2020). Here we investigated condensable oxygenated organic
vapors observed by nitrate CI-APi-TOF in August-September 2019 at the Station for
Observing Regional Processes and the Earth System (SORPES) in the western part of
the YRD, an anthropogenic-emissions-dominated environment (Fu et al., 2013;Xu et
al., 2017) mixed with enhanced biogenic emissions during summer (Wang et al.,
2020a;Xu et al., 2021). A variety of oxidants (Liu et al., 2019;Li et al., 2020;Xia et al.,
2020) with numerous precursors (VOCs) suggest very complicated atmospheric
oxidation processes and thousands of products (OOMs). Thereby, positive matrix
factorization (PMF) (Paatero and Tapper, 1994) was applied to time-resolved mass
spectra which had been pre-divided into small bins (binPMF, Zhang et al., 2019), to





separate various sources or processes of OOMs. Combined with summarizing the
ensemble chemical characteristics of OOMs, some interesting inspirations about the
conversion of VOCs to OOMs were obtained.

**2 Methodology**

**2.1 Study site**

The SORPES station (32°07′14″ N, 118°57′10″ E; 62 m a.s.l.) is located at Nanjing in
the western part of YRD, one of the most developed regions in eastern China. Due to
its unique location, this site can be influenced by air masses from different source
regions of anthropogenic emissions, biomass burning, dust and biogenic emissions
(Ding et al., 2013;Ding et al., 2016). Detailed descriptions for the station can be found
in previous studies (Nie et al., 2015;Xie et al., 2015;Xu et al., 2018;Wang et al.,
2018a;Sun et al., 2018;Shen et al., 2018).

**2.2 Instrumentation**

The nitrate CI-APi-TOF (Aerodyne Research Inc. and Tofwerk AG), combining a
chemical ionization source (CI) and an atmospheric pressure interface time-of-flight
mass spectrometer (APi-TOF) equipped with a long-TOF model (LTOF) with mass
resolution of 8000-12000 Th/Th, was deployed to detect the ambient sulfuric acid and
OOMs. The ambient air was pulled into a laminar flow reactor, where the sample flow
(10 L min$^{-1}$) is surrounded by a purified airflow serving as the sheath flow (25 L min$^{-1}$
$^{1}$), through a stainless-steel tube (a 100 cm long, 3/4 in. diameter). Nitrate reagent ions
were generated in the sheath flow by exposing air-containing nitric acid to a
PhotoIonizer X-Ray (Model L9491, Hamamatsu, Japan). Detailed description of the
instrument has been described elsewhere (Junninen et al., 2010;Jokinen et al., 2012).
The data were acquired at 1 Hz time resolution and analyzed with a tofTools package
(version 6.11) based on MATLAB (MathWorks Inc.). The quantification of OOMs was
derived via Eq. (1).

$$[\text{OOM}_i] = \ln\left(1 + \frac{\sum_{n=0}^{1}[\text{OOM}_i \cdot (\text{HNO}_3)_n \cdot \text{NO}_3^- + (\text{OOM}_i - \text{H})^-]}{\sum_{n=0}^{2}[(\text{HNO}_3)_n \cdot \text{NO}_3^-]}\right) \times C \times T_i \tag{1}$$

Here [OOM$_i$] is the concentration (molecules cm$^{-3}$) of the OOM molecule. On the right
side of the equation, the numerator in the parenthesis is the observed total signals (ions/s)
of OOM ions charged in different way, the denominator is the sum of all reagent ion
signals (ions/s). First, a H$_2$SO$_4$-based calibration factor C (4.2×10$^9$ molecules cm$^{-3}$) was
inferred from a calibration using H$_2$SO$_4$ (Kuerten et al., 2012) proceeding taking into
account the diffusion loss in the sampling line, by assuming that all detected OOMs
have the same ionization efficiency as H$_2$SO$_4$ and that the (OOM·NO$_3$-) clusters are
very stable without dissociation during their residence time of detection. Second, a mass
dependent transmission efficiency T$_i$ of APi-TOF was inferred in a separate experiment
by depleting the reagent ions with several perfluorinated acids (Heinritzi et al., 2016).






VOCs precursors were measured by a proton transfer reaction time-of-flight mass
spectrometer (PTR-ToF-MS, Ionicon Analytik, Innsbruck, Austria, TOF 1000 ultra).
PM$_{2.5}$ was measured with a combined technique of light scattering photometry and beta
radiation attenuation (Thermo Scientific SHARP Monitor Model 5030). The chemical
compositions of PM$_{2.5}$ was determined on-line using time-of-flight aerosol chemical
speciation monitor (TOF-ACSM, Aerodyne Research Inc.). PMF analysis was further
used to separate the organic aerosol (OA) to primary and secondary organic aerosols
(POA and SOA). The number concentrations of particles were measured by the
scanning mobility particle sizer (SMPS) with nano DMA (4.0 to 63.8 nm) and long
DMA (41.4 to 495.8 nm) and the aerodynamic particle sizer (APS) (0.5 to 18.0 μm).
NO and NO$_2$ were measured using a chemiluminescence analyzer equipped with a blue-
light converter (TEI, Model 42I-TL); O$_3$, SO$_2$, and CO were measured using the
ultraviolet photometry, pulsed-UV fluorescence, and IR (infrared) photometry
techniques (TEI, Model 49I, 43C, and 48C), respectively. Zero and span calibrations
for trace gases were performed weekly during the campaign. Meteorological
measurements including relative humidity (RH), wind speed, wind direction, and air
temperature were recorded by Automatic Weather Station (CAMPEEL co., AG1000).
J(O$^1$D) was measured by ultra-fast CCD-detector spectrometer, UVB enhanced
(Meteorologieconsult Gmbh, Germany).

**2.3 Hydroxyl radical (OH) estimate**
The OH concentration was calculated by applying the Eq. (2), based on the assumption
that gaseous sulfuric acid (SA) is mostly produced from the oxidation of SO$_2$ by OH
and primarily loss by condensing onto particles, denoted as condensation sink (CS). It
has been proved that $\frac{k_{OH+SO_2}\cdot[SO_2]\cdot[\cdot OH]}{CS}$ is a very reliable proxy for SA during the day
(Lu et al., 2019). The ozonolysis of alkenes can form stabilized Criegee intermediates
(SCIs) in addition to OH, and SCIs can also oxidize SO$_2$ to form SA (Mauldin Iii et al.,
2012;Guo et al., 2021). A previous study on SA proxy in this site has revealed that the
reactions of SO$_2$ with products from the ozonolysis of alkenes generate a moderate
amount of nighttime sulfuric acid, with little effect on daytime sulfuric acid (Yang et
al., 2021). Thus, OH may be overestimated during nighttime. In this study, OH was
used to calculate the production rates of RO$_2$ (Fig. 4), the error of OH do not change
the relative distribution of RO$_2$ from different precursors. And OH is mainly used when
analyzing daytime data.

$$[OH] = \frac{SA \cdot CS}{k_{OH+SO_2}\cdot[SO_2]} \qquad (2)$$


**2.4 binPMF**





binPMF has been used to analyze the measured HR mass spectrometry data. Briefly,
the raw spectra with were divided into narrow bins with a width of 0.006 Th after mass
calibration. The data matrix and error matrix were prepared according to the methods
described by Zhang et al. (2019) for the PMF model inputs (Section S1 in the
supplement). Different from the traditional PMF such as using unit mass resolution
(UMR) or HR data as input, binPMF still retains HR information as much as possible,
avoids the uncertainty of HR peak fitting influencing the results of PMF, and separate
the complex overlapping peaks for fitting. The PMF analysis in this work uses the
IGOR based analyzing interface SoFi (solution finder, version 6.8) and ME-2 as
described in (Canonaco et al., 2013). After select the PMF solution, we fitted the HR
peaks in each factor through toftool.
**3 Results and Discussions**
Figure 1 shows temporal variation of OOMs and related parameters at the SORPES
station in the northeastern suburb of Nanjing from August 02 to September 06, 2019.
During the observation period, 22 of 35 days had maximum hourly temperatures above
30 degrees Celsius, and 29 days had maximum hourly $J(O^1D)$ above $2\times10^{-5}$ s$^{-1}$. High
temperature and solar radiation indicate strong photochemistry, producing a large
amount of ozone, with concentration often exceeding 80 ppb. Even at night, the
concentration of ozone is rarely lower than 10 ppb, resulting from the weak titration of
low NO. At the same time, the reaction between ozone and high concentration of $NO_2$
can provide sufficient $NO_3$ radicals, dominating nocturnal degradation of certain
volatile organic compounds (VOCs) (Wayne et al., 1991). The elevated mixing level of
total aromatic hydrocarbons is one of the main characteristics of the atmosphere in
densely populated areas, in addition to which there should be many alkanes and alkenes
which cannot be observed by PTR-ToF-MS (Fu et al., 2013;Xu et al., 2017). In the
daytime with strong photochemical reaction ($J(O^1D)> 1\times10^{-5}$ s$^{-1}$), we instead observed
higher concentrations of isoprene than total aromatics ([isoprene]$_{median}$=1.3 ppb,
[aromatics]$_{median}$=1.1 ppb). The complex mixtures of anthropogenic and biogenic VOCs
can be oxidized through a variety of pathways to produce OOMs, of which some low
volatile components will condense into particles, forming organic aerosol. The
concentrations of OOMs with mass-to-charge ratio (m/z) below 360 Th are usually
higher than $10^6$ molecules cm$^{-3}$, and some can even reach up to $10^7$-$10^8$ molecules cm$^{-}$
$^3$. Clustered peaks on the spectra of OOMs and their clear daily variations imply a lot
of chemical and physical dynamics information (Fig. 1(d), see Fig. S1(a) for normalized
spectra), which is the main aspect we want to explore in this work.



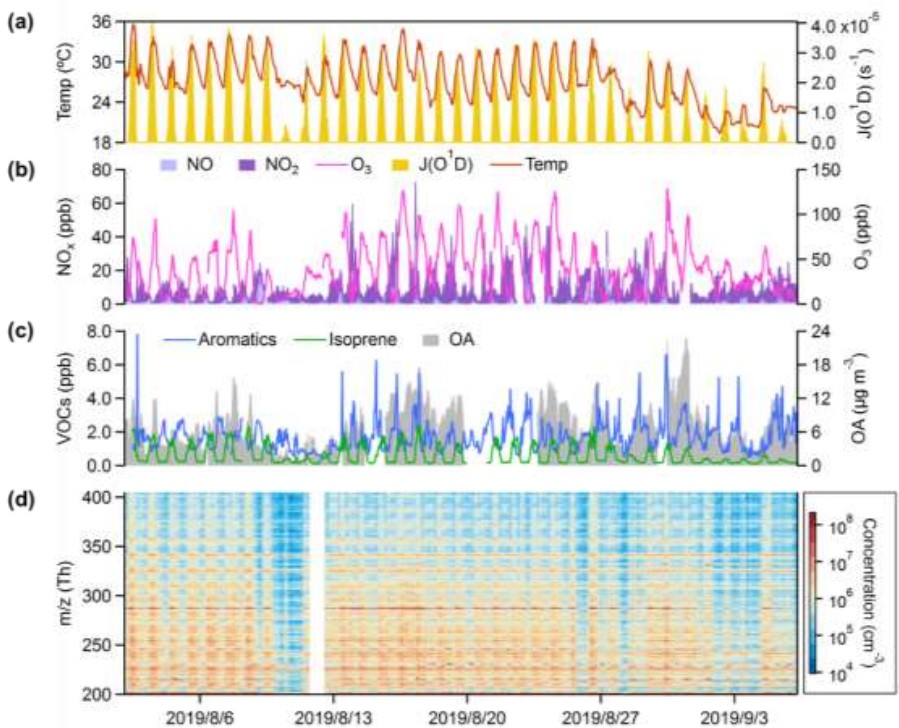

Figure 1. Overview of the observation. Time series of (a) temperature (Temp) and the photolysis frequency of $O_3$ ($JO^1D$), (b) $O_3$ and $NO_x$ ($NO+NO_2$), (c) total aromatics (benzene + toluene + $C_8$ aromatics + $C_9$ aromatics + $C_{10}$ aromatics + styrene), isoprene, and OA, and (d) mass spectra of nitrate CI-APi-TOF with m/z in the range of 202-404 Th.

The binPMF analysis was performed to characterize the sources or processes of OOMs. A 14-factor solution was selected to interpret the data set, including 3 factors about AVOCs daytime chemistry, 3 isoprene-related factors, 3 factors about BVOCs nighttime chemistry, 3 factors about nitrated phenols (NP), and 2 factors excluded from the following discussion. One of these two disregarded factors is mainly composed of fluorinated contaminations (F-contaminations), and the other is mainly a mixture of nitrated phenols and fluorinated contaminations (Mixed contaminations). When naming these factors, we prioritize the description of dominated species or their precursors, but if the precursors are complex mixtures, our naming highlights the characteristics of the chemical processes that drive certain factors. Although this may not be the optimal PMF solution, it still separates a lot of useful information. We also stress that the urban OOMs mix is unlikely to be a perfect combination of independent, unchanging factors, which is an underlying assumption in the PMF algorithm. As such, there will be no solution which is complete and perfect, but we chose a solution from which we were able to provide us with interesting insights. Details of the PMF diagnostics is provided in section S1 in the supplement. For the convenience of





discussions, we have grouped these factors. The factors in each group follow a similar
but not entirely exact pattern.

Table 1. Summary of molecular characteristics of 9 discussed non-nitrated-phenols
factors. The calculation of the relevant parameters is given in section S2 in
the supplement. Major peaks of each factor are summarized in section S3 in
the supplement.

| Factor | Average concentration (cm-3) | Effective formulae | MW (g mol$^{-1}$) | OSc | O:C | N:C | DBE | $\log_{10}(C^*(\mu g\ m^{-3}))$ in 300K |
|---|---|---|---|---|---|---|---|---|
| Aro-OOMs | 1.86E+07 | $C_{9.1}H_{14.3}O_{6.1}N_{0.6}$ | 230.2 | -0.52 | 0.73 | 0.08 | 2.6 | -1.7 |
| Temp-related | 4.50E+07 | $C_{6.8}H_{10.2}O_{6.0}N_{0.5}$ | 195.8 | -0.02 | 0.95 | 0.08 | 2.5 | -1.4 |
| Aliph-OOMs | 2.11E+07 | $C_{7.5}H_{12.2}O_{6.7}N_{1.2}$ | 225.7 | -0.55 | 0.96 | 0.17 | 1.9 | 0.0 |
| Photo-related | 4.77E+07 | $C_{6.9}H_{11.0}O_{7.4}N_{1.2}$ | 228.3 | -0.28 | 1.18 | 0.20 | 1.8 | -1.1 |
| $O_x$ & SOA-related | 2.59E+07 | $C_{6.6}H_{9.8}O_{6.8}N_{1.1}$ | 214.2 | -0.24 | 1.11 | 0.19 | 2.2 | -0.3 |
| Isop-OOMs | 2.83E+07 | $C_{5.5}H_{9.6}O_{6.9}N_{1.4}$ | 205.8 | -0.51 | 1.34 | 0.28 | 0.9 | 1.2 |
| BVOCs-OOMs I | 1.68E+07 | $C_{7.2}H_{11.5}O_{7.0}N_{1.0}$ | 224.1 | -0.26 | 1.06 | 0.16 | 2.0 | -1.4 |
| BVOCs-OOMs II | 9.05E+06 | $C_{9.2}H_{14.6}O_{7.1}N_{0.9}$ | 251.3 | -0.45 | 0.83 | 0.11 | 2.5 | -2.8 |
| BVOCs-OOMs III | 1.57E+07 | $C_{8.6}H_{13.7}O_{6.9}N_{1.2}$ | 243.3 | -0.64 | 0.87 | 0.16 | 2.1 | -0.7 |

Note: MW is the molecular weight, OSc is the carbon oxidation state, O:C is the oxygen
to carbon ratio, N:C is the nitrogen to carbon ratio, DBE is the double bond equivalent,
C* the saturation concentration and $\log_{10}(C^*)$ is the volatility.

## 3.1 AVOCs daytime chemistry

The following daytime factors are characterized by $C_6$-$C_9$ OOMs (Fig. 2(a)), considered
to be derived from the oxidation of anthropogenic VOCs in this urban atmosphere,
while we cannot completely exclude the present of BVOCs-derived OOMs, such as $C_5$
and $C_{10}$ OOMs.



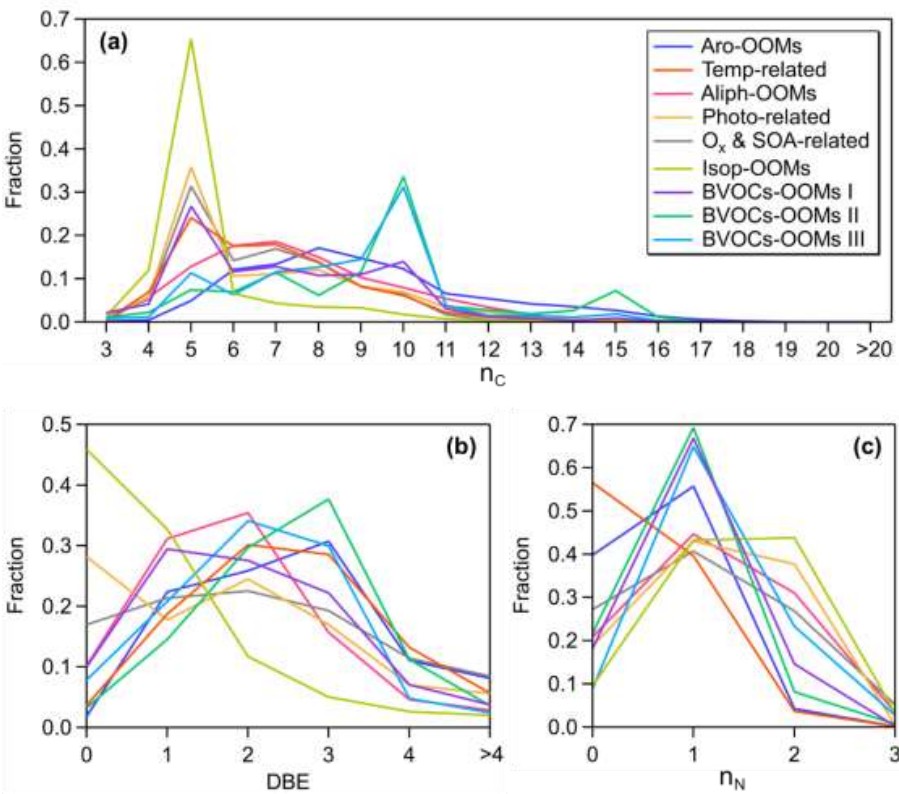

Figure 2. The distributions of observed non-nitro OOMs grouped by (a) the number of
carbon atoms ($n_C$), (b) DBE, and (c) the number of nitrogen atoms ($n_N$) in 9 factors.
Since the signals of $RO_2$ are very weak, $RO_2$ from BVOCs OOMs I and BVOCs OOMs
II are excluded in (b) to keep the integer value of DBE.
**Aro-OOMs factor**
The effective DBE of this factor is the largest among all factors (Table 1), with main
signals come from compounds with DBE > 2 (Fig. 2(b)) and consistent with the nature
of the oxidation products of aromatics (Fig. 3(a)). Combined with the correlation with
the production rates of OH-initiated primary peroxy radicals ($RO_2$) from aromatics
calculated by Eq. (3) (Fig. 4), this factor is supposedly dominated by aromatics-derived
OOMs (Aro-OOMs). The Aro-OOMs factor increases from 5:00 LT with a maximum
at 10:00 LT and a sub peak around 16:00 LT (Fig. 3(e)), following the diurnal variations
of $P_{RO2}$ of $C_7$-$C_{10}$ aromatics (Fig. 4(b-d)) but poorly correlated with $P_{RO2}$ of benzene
(Fig. 4(a)). Furthermore, OOMs with 8 carbon atoms have the highest signal in this
factor (Fig. 2(a)), derived from the most abundant $C_8$-aromatics + styrene $RO_2$ (Fig.
4(f)). Both of these can be explained by the fact that substituted aromatics have higher
OH reactivity (Bloss et al., 2005) and higher HOM yields (Wang et al., 2017;Molteni
et al., 2018) than their homologues with less carbon atoms. In terms of molecular
formula, the aromatics-derived OOMs have an overlap with monoterpenes-derived
OOMs (Mehra et al., 2020). Monoterpenes can contribute more $C_{10}$ OOMs than
aromatics ($P_{MT-RO_2} > P_{C_{10} Aro-RO_2}$), but aromatics play a more important role in total
in this factor since they provide more $RO_2$ in the urban atmosphere (Fig. 3(f)).
$$P_{RO_2} = k_{OH+VOC} \bullet [OH] \bullet [VOC] \qquad (3)$$

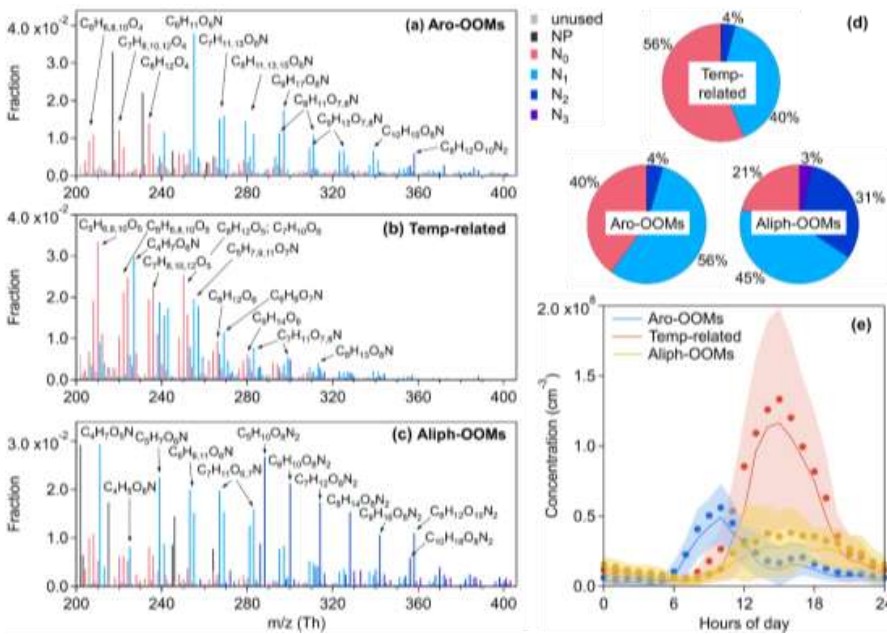

Figure 3. Mass spectra of (a) the Aro-OOMs factor, (b) the Temp-related factor, (c) the
Aliph-OOMs factor, and the elemental formulas of major peaks are labeled above them.
Peaks are color-coded by $n_N$ as indicated at the top right of the figure, and the fractions
of peaks grouped by $n_N$ are reported in (d) the pie chart. The gray sticks are fluorinated
contaminations, or non-identified compounds. The nitrated phenols are drawn
separately with black peaks in (a), (b) and (c), and were not included in (d). So $n_N$ can
more reliably represent the number of nitrate groups in each molecule. Diurnal patterns
(Beijing time) of these three factors are shown in (e), the bold solid lines are the median
values, shaded areas represent percentiles of 75 % and 25 % and solid circles represent
mean values.
The main molecules of the Aro-OOMs factor are summarized in Table S2. The $C_xH_{2x-5}O_6N$ (x=6-12, of which $C_8H_{11}O_6N$ is the most intense) series can be produced by the
reaction (R1a) of NO with the bicyclic peroxy radicals (HO-Ar-$(O_2)_2$), the key
intermediates for aromatics oxidation proposed in the MCM (Bloss et al., 2005;Birdsall
and Elrod, 2011). And here dihydroxy nitro-BTEX ($C_xH_{2x-7}O_4N$, x=6-8) can be treated





as indicators of aromatics oxidation. In addition to the conventional products, $C_9H_{13}O_{7-9}N$ from the $C_xH_{2x-5}O_{7-9}N$ (x=7-13) series are also significant in the OH-initiated and
$_9N$ from the $C_xH_{2x-5}O_{7-9}N$ (x=7-13) series are also significant in the OH-initiated and
$NO_x$-influenced oxidation experiments of 1,2,4-trimethylbenzene    (Zaytsev et al.,
2019) and of 1,3,5-trimethylbenzene (Tsiligiannis et al., 2019). More oxygenated
compounds may come from auto-oxidation and multigenerational OH attacks. However,
the effective OSc of this factor (Table 1) is lower than that of oxidation products of
aromatics in recent laboratories (Zaytsev et al., 2019;Tsiligiannis et al., 2019;Garmash
et al., 2020;Wang et al., 2020c). We speculate that the abundances of $NO_x$ relative to
oxidants and precursors in these experiments are not sufficient to reproduce the
atmospheric conditions during our observation, or that HOMs are more concentrated in
aerosols due to the large condensation sink on this site (Qi et al., 2015). Although
species with DBE < 3 (Fig. 2(b)) in this factor are most likely produced from multiple
OH attacks in aromatics oxidation, we can't rule out the contribution of alkanes co-
emitted with aromatics, such as the series $C_xH_{2x-1}O_6N$ (x=5-14).

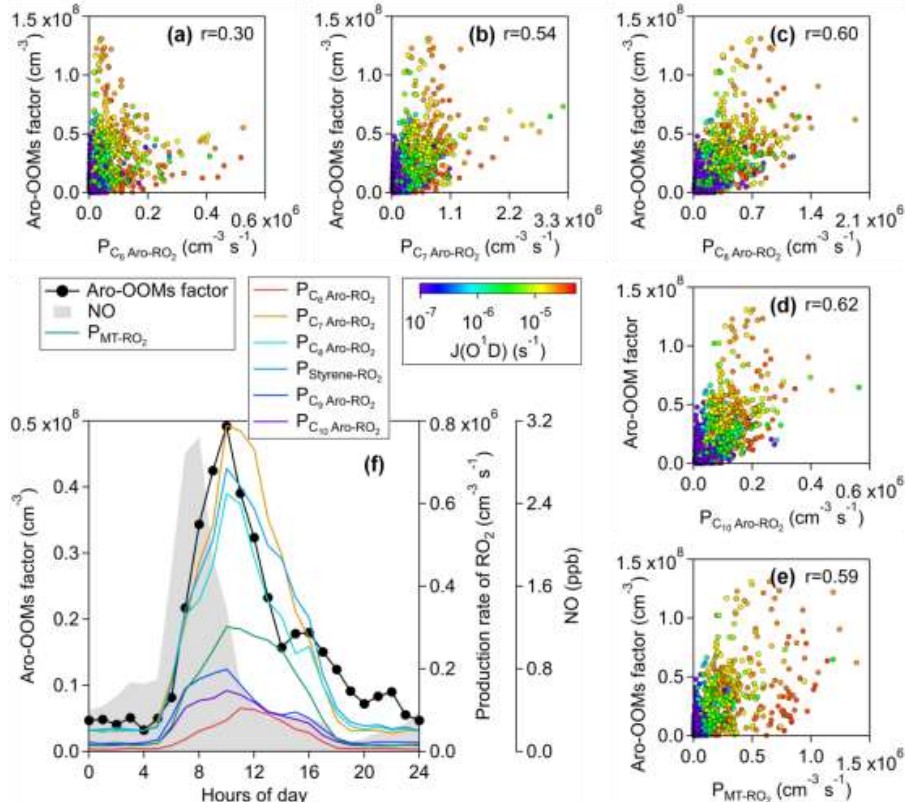

Figure 4. Correlations of the Aro-OOMs dominated factor with production rate of $RO_2$
from OH-initiated oxidation of (a) benzene ($P_{C_6\ Aro-RO_2}$), (b) toluene ($P_{C_7\ Aro-RO_2}$), (c)
$C_8$ aromatics ($P_{C_8\ Aro-RO_2}$), (d) $C_{10}$ aromatics ($P_{C_{10}\ Aro-RO_2}$), and (e) Monoterpenes


($P_{MT-RO_2}$). All the scatters are colored by J(O$^1$D), to show the difference between day
and night. The median diurnal patterns of this factor and related parameters are plotted
in (f).

**Temp-related factor**

This factor is named due to good correlation with temperature (Fig. 5), and shows
maximum intensity in the afternoon around 15:00 (Fig. 3(e)). The Temp-related factor
is the only one dominated by non-nitrogenous organics (Fig. 3(b) and (d)), and has the
highest effective OSc (Table 1) among all the factors. The $C_xH_{2x-4}O_5$ (x=5-11,
summarized in Table S3), $C_xH_{2x-2}O_5$ (x=5-10), $C_xH_{2x-6}O_5$ (x=5-11), and $C_xH_{2x-4}O_6$
(x=5-10) series are possibly products from RO$_2$ terminated by HO$_2$ (R2a), or closed-
shell products from RO in reactions R3a or R3b. Temperature starts to rise at 6:00 LT
(Fig. 12(b)), but this factor does not accumulate significantly until after about 10:00 LT
(Fig. 3(e)), when the mixed level of NO is reduced to 1 ppb (Fig. 4(f)). This
phenomenon suggests a probability of HO$_2$-driven chemistry of this factor under low
NO conditions, since that NO can consume HO$_2$ and compete with HO$_2$ for RO$_2$. Such
low-NO atmospheric oxidation pathways have been suggested to be non-negligible in
the afternoon in central Beijing (Newland et al., 2021).

A factor caused by similar chemical processes called isoprene afternoon was discovered
in the nitrate CI-APi-TOF data collected at a forest site in Centreville, Alabama, USA
(Massoli et al., 2018), correlated well with HO$_2$, O$_3$, and temperature. We also observed
a number of isoprene oxidation products in the Temp-related factor (nC = 4, 5 in Fig.
2(a)). Many of the $C_xH_{2x-1}O_6N$ (x=3-7) and $C_xH_{2x-3}O_6N$ (x=4-9) series were also present
in the light HOMs factor which was supposed to be fragments from the oxidation of
larger VOCs (e.g., monoterpene) in Hyytiälä, Finland (Yan et al., 2016). While at the
SORPES station, the C$_6$-C$_9$ ions should mainly come from the oxidation of
anthropogenic VOCs. At lower temperatures, the propensity of condensable organic
molecules to condense into aerosol makes the concentration measured using nitrate CI-
APi-TOF lower. Thus, the total concentration of the Temp-related factor in the gas and
aerosol phases was calculated based on gas-particle equilibrium (section S4 in
the supplement), and was found to be still temperature dependent (Fig. S6), illustrating
the temperature-influenced chemical process controlling the factor. For instance,
Unimolecular reaction rates like RO$_2$ H-shifts increase qualitatively with temperature
(Bianchi et al., 2019;Frege et al., 2018).

$$RO_2\bullet \ + \ NO \ \rightarrow \ RONO_2 \qquad\qquad (R1a)$$
$$RO_2\bullet \ + \ NO \ \rightarrow \ RO\bullet \ + \ NO_2 \qquad\qquad (R1b)$$
$$RO_2\bullet \ + \ HO_2\bullet \ \rightarrow \ ROOH + \ O_2 \qquad\qquad (R2a)$$
$$RO_2\bullet \ + \ HO_2\bullet \ \rightarrow \ RO\bullet \ + \bullet OH + \ O_2 \qquad\qquad (R2b)$$
$$RO\bullet \ + \ O_2 \ \rightarrow \ RC=O \ + \ HO_2\bullet \qquad\qquad (R3a)$$
$$RO\bullet \ + \ O_2 \ \rightarrow \ fragments \qquad\qquad (R3b)$$




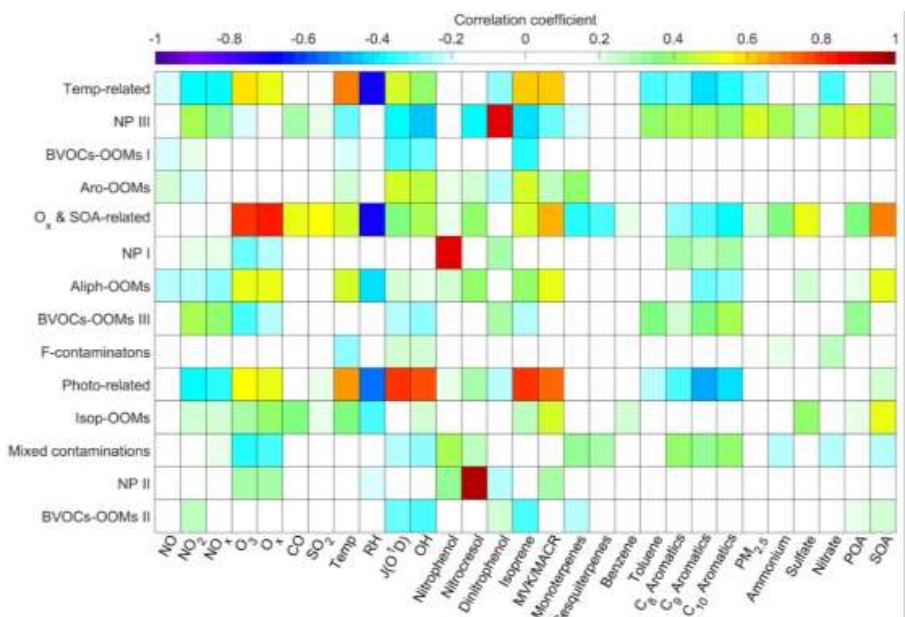


Figure 5. Correlations of PMF factors with external gas-phase and particulate tracers
from other instruments deployed at the SOPRES station, with the color representing the
Pearson correlation coefficients. From left to right, the tracers are gas-phase species
(NO, $NO_2$, $NO_x$, $O_3$, CO, $SO_2$), meteorological data (temperature (Temp), relative
humidity (RH), photolysis constants ($J(O^1D)$)), nitrate CI-APi-TOF data (OH,
nitrophenol, nitrocresol, dinitrophenol), PTR-ToF-MS data (isoprene, methyl vinyl
ketone/methacrolein (MVK/MACR)), monoterpenes, sesquiterpenes, benzene, toluene,
$C_8$ aromatics, $C_9$ aromatics, $C_{10}$ aromatics), PM$_{2.5}$, and ACSM data (ammonium, sulfate,
nitrate, POA, SOA).

**Aliph-OOMs factor**

This factor is dominated by organic nitrates (Fig. 3(c) and (d)), and contains the bulk
of anthropogenic di-nitrates and tri-nitrates. The $C_xH_{2x-2}O_8N_2$ (x=4-13, summarized in
Table S3) and $C_xH_{2x}O_8N_2$ (x=4-9) series have not been reported in aromatics oxidation
experiments under high $NO_x$ conditions (Tsiligiannis et al., 2019;Wang et al., 2020c),
and nor in the forest or rural environments (Yan et al., 2016;Massoli et al., 2018). A
reasonable assumption is that these saturated or nearly saturated compounds are the
products of aliphatics (including alkanes, alkenes, aliphatic alcohol, etc.) during their
oxidation affected intensively by $NO_x$ in the urban atmosphere. The Aliph-OOMs factor
has a broad afternoon peak lasting from 14:00 to 19:00 LT (Fig. 3(e)), suggesting that
the formation of multi-nitrate requires enough OH exposure time.

Considering a simple scenario of alkane photo-oxidation under high $NO_x$ conditions:





the $RO_2$ generated from OH attack is completely terminated by NO (Fig. 6(a)). The
chain-retaining products are $C_nH_{2n}O$ (one more carbonyl group than the precursor) and
$C_nH_{2n+1}O_3N$ (one more nitrate group than the precursor), and the re-oxidation of these
products is a repetition of the above process which is defined as the basic reaction
scheme. The multiple (1st to 3rd) generation products of alkanes summarized in Fig.
6(b) are regarded as reference compounds, which we compare OOMs with to
investigate other mechanisms that differ from those shown in Fig. 6(a). Specifically,
this comparison is performed between the reference molecule and OOMs with the same
numbers of carbon, hydrogen and nitrogen atoms, but different numbers of oxygen
atoms. The number of extra oxygen ($n_{O_{extra}}$) from each aliphatic OOM over its
corresponding reference molecule was calculated by Eq. (4), that is, subtracting
carbonyl and nitrate oxygens from the molecule. Thus, the $n_{O_{extra}}$ can represent the
additional oxygenated moieties such as hydroxyl group (-OH), peroxy group (-OOH),
and possibly ether group. These functional groups may come from RO isomerization
(Orlando et al., 2003), the addition of OH to alkenes, or pre-existing moieties in the
precursor, $RO_2$ autoxidation or specific $RO_2$ bimolecular termination reactions
($RO_2+RO_2$, $RO_2+HO_2$).
$$n_{O_{extra}} = n_O - DBE - 3 \times n_N \qquad (4)$$
As showed in Fig. 6(c), aliphatic OOMs in this factor are mainly the third-generation
products followed by the second-generation products, and both of which have one or
two oxygen-containing functional groups in addition to the carbonyls and nitrates. It
should be noted that the first-generation (Fig. 6(a)) and basic products (Fig. 6(b)) here
are underestimated due to the low sensitivity of nitrate CI-APi-TOF to these compounds.
The multifunctional products of aliphatics are condensable to form SOA (correlation
coefficients with SOA showed in Fig. 5). Recent work has showed that autoxidation is
more common than previously thought (Wang et al., 2021), and more studies are needed
to explore the oxidation mechanisms of anthropogenic aliphatics and to evaluate their
contribution to SOA.









Figure 6. (a) Simplified oxidation mechanism for alkanes attacked by OH once under
NOₓ-controlled conditions. (b) summarizes the changes in molecular formula of the 1st
to 3rd generation products of alkanes, based on the basic reaction scheme in (a). (c)
shows the fraction of potential alkanes-derived compounds in the Aliph-OOMs factor.
The compounds listed in (c) are grouped according to the molecular formulas in (b),
i.e., the same number of carbon, hydrogen and nitrogen atoms, but different numbers
of oxygen atoms. The bars in (c) are colored with $n_{O_{extra}}$. Please see text for details
about $n_{O_{extra}}$.

### 3.2 Isoprene-related chemistry

The following factors are characterized by $C_5$ OOMs (Fig. 2(a)), of which an isoprene
dihydroxyl dinitrate $C_5H_{10}O_8N_2$ (charged by $NO_3^-$ at m/z 288 Th) is the fingerprint
molecule (Fig. 7). Apart from isoprene-derived compounds, OOMs formed from other
precursors undergoing the similar chemical processes are also allocated to these three
factors.

### Photo-related factor

This factor is defined based on its correlation with $J(O^1D)$ (Fig. 5), having an apparent
diurnal cycle with a peak at 12:00 LT (Fig. 7(e)). The major peak of the Photo-related
factor is $C_5H_{10}O_8N_2$ (Fig. 7(a)), most probably generating from double OH attack
proceed with double $RO_2$+NO termination (Jenkin et al., 2015). $C_5H_{10}O_8N_2$ can be also
produced in $NO_3$+ isoprene system (Ng et al., 2008;Zhao et al., 2020), whereas in this
study, the nocturnal $C_5H_{10}O_8N_2$ is principally from the Isop-OOMs factor (Fig. 8(b))
which will be discussed later. Other peaks with nC≤5, like $C_5H_7O_7N$, $C_4H_7O_6N$,
$C_5H_9O_6N$, are also likely to be the isoprene products. The total signal of compounds
with nC > 5 is not low, although their respective proportions are not as prominent as $C_5$
species (Fig. 7(d)), implying the contribution of other precursors together with isoprene.
In addition, the relationship of this factor with isoprene and $J(O^1D)$ together (Fig. 5)
reveals the effect of light-dependent emission of isoprene on it.

### Oₓ & SOA-related factor

The atmospheric oxidation of VOCs produces low-volatile compounds, forming SOA
through gas-particle partitioning, and concurrently promotes ozone formation
(Atkinson, 2000). Both SOA and $O_x$ have long lifetimes (>12 h), and their correlations
have been extensively investigated (Herndon et al., 2008;Wood et al., 2010;Hu et al.,
2016). The OOMs factor related to ozone and SOA together (Fig. 5), having slightly
elevated concentrations during daytime (Fig. 7(e)), is considered to be generated from
this photochemical aging process. Apart from $C_5H_{10}O_8N_2$, other isoprene multi-nitrates
are also present in this factor. $C_5H_9O_{10}N_3$, an isoprene hydroxyl trinitrate requiring at



least two steps of oxidation found in the experimental study on isoprene oxidation by
NO$_3$ (Zhao et al., 2020), naturally does not appear in the photo-related factor at all, but
is mostly apportioned into the O$_x$ & SOA-related factor and the Isop-OOMs factor (Fig.
8(c) and 8(d)). Like the photo-related factor, isoprene is a significant but not the only
precursor of this factor (Fig. 2 and 7). The biggest peak of the O$_x$ & SOA-related factor
is an ion at m/z 264 with formula C$_6$H$_5$O$_3$N (HNO$_3$NO$_3^-$), identified as an adduct of
nitrophenol (C$_6$H$_5$O$_3$N) with nitrate dimmer (HNO$_3$NO$_3^-$). The time variation of
C$_6$H$_5$O$_3$N (HNO$_3$NO$_3^-$) is influenced by the reagent ions in addition to the atmospheric
nitrophenol. So far, we don't know why this compound share the same processes with
others, but we did a test that removing the bins with unit m/z = 264 from the input
matrix and still got this factor from PMF model.

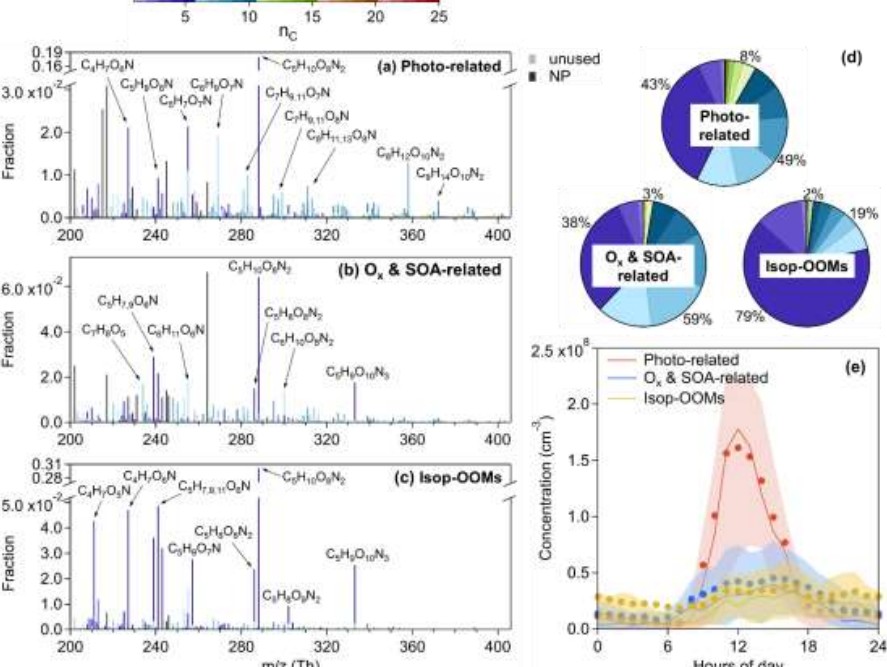

Figure 7. Mass spectra of (a) the Photo-related factor, (b) the O$_x$ & SOA-related factor,
(c) the Isop-OOMs factor, and the elemental formulas of major peaks are labeled above
them. Peaks are color-coded by n$_C$ as indicated at the top of the figure, and the fractions
of peaks grouped by n$_C$ are reported in (d) the pie chart. The gray sticks are fluorinated
contaminations, or non-identified compounds. The nitrated phenols are drawn
separately with black peaks in (a), (b) and (c). The molecules represented by the gray
and black sticks were not included in (d). Diurnal patterns of the three factors are shown
in (e), the bold solid lines are the median values, shaded areas represent percentiles of
75 % and 25 % and solid circles represent mean values.
**Isop-OOMs factor**






The mass spectra of the Isop-OOMs factor, as its name implies, is exclusively
contributed by isoprene-derived compounds (Fig. 7(c)). $C_5H_{10}N_2O_8$ contributes about
30% of the intensity of this factor, and the dominance of $C_5H_{10}N_2O_8$ was also found in
the isoprene nitrates type I factor in Centreville (Massoli et al., 2018). In addition to
multi-nitrates ($C_5H_{10}O_{7-8}N_2$, $C_5H_8O_{6-9}N_2$, and $C_5H_9O_{10}N_3$ summarized in Table. S6),
several mononitrate series ($C_4H_7O_{5-7}N$, $C_5H_9O_{4-9}N$, $C_5H_7O_{5-8}N$, and $C_5H_{11}O_{5-6}N$) of
this factor are also abundant in the isoprene nitrates type II factor in Centreville
(Massoli et al., 2018). Many of isoprene nitrates here have been specially investigated
in our previous observations in the YRD (Xu et al., 2021), and have been discovered in
other filed measurements (Lee et al., 2016;Massoli et al., 2018) and in many
laboratories (Ng et al., 2008;Lambe et al., 2017). Generally, these compounds are
second- and third-generation OH oxidation products of isoprene under high-$NO_x$
conditions (Wennberg et al., 2018).

The diurnal pattern of the Isop-OOMs factor is relatively unclear (Fig. 7(e)), with
obvious differences between mean and median values usually caused by plume events.
This indicates that isoprene chemistry, usually varying evidently from day (OH-
initiated) to night ($NO_3$-initiated), is not the driver of this factor. This factor correlates
positively with MVK / MACR and SOA (r>0.50, showed in Fig. 5), but not with
isoprene and OH. It seems that these isoprene OOMs are produced elsewhere and then
transported due to their longer lifetime determined by their relatively high volatility
(Table 1). The Isop-OOMs factor in the continental air masses are more intensive than
those in the coastal and YRD air masses (Fig. S7), consistent with the spatial
distribution of isoprene emissions (Sindelarova et al., 2014). An archetypal episode
affected by continental air masses (August 13 to August 17, 2019) is showed in Fig. 8.
During this period, $C_5H_9O_{10}N_3$ was almost entirely transported, while $C_5H_{10}O_8N_2$ has
strong in situ photochemical generation, in addition to the source of transport.

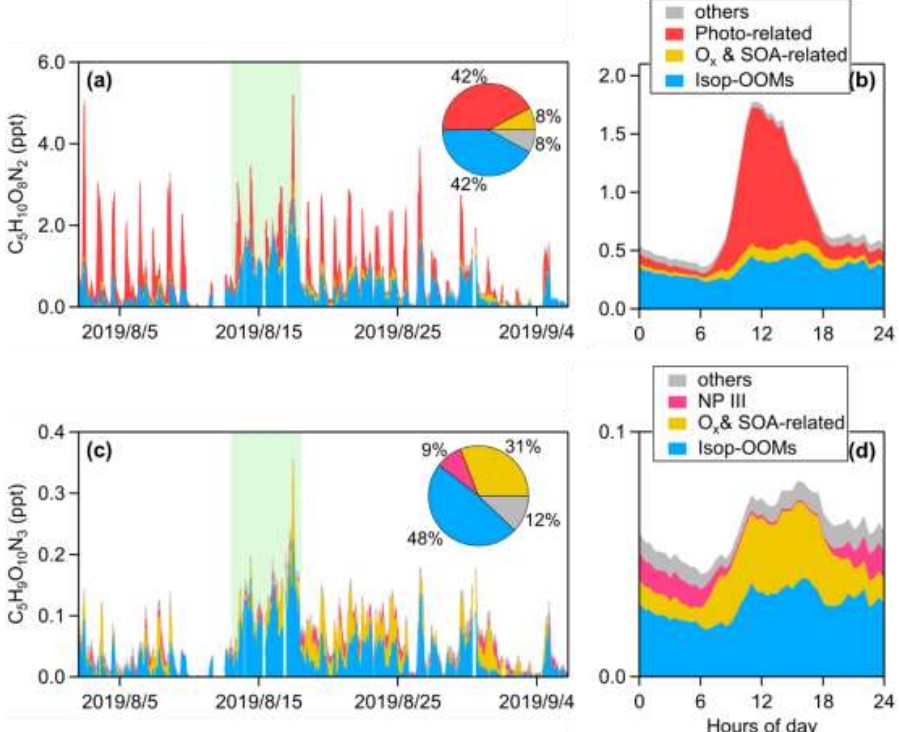


Figure 8. Stacked (a) time series and (b) mean diurnal pattern of isoprene dihydroxyl
dinitrate ($C_5H_{10}O_8N_2$). Stacked (c) time series and (d) mean diurnal pattern of isoprene
hydroxyl trinitrate ($C_5H_9O_{10}N_3$). The contribution ratios of each PMF factor to these
two compounds are reported in the pie chart respectively. The light green shaded area
represents a typical episode influenced by transported continental air masses (August
13 to August 17, 2019).

**3.3 BVOCs nighttime chemistry**

The following nighttime factors are characterized by $C_{10}$ OOMs (Fig. 2(a)), which are
identified as the oxidation products of monoterpenes. Except for the BVOCs-OOMs I
factor (Fig. 9(a)), the contribution of isoprene-derived OOMs was much lower in these
factors. Compared to the above isoprene-related factors, $C_5H_{10}O_8N_2$ and $C_5H_9O_{10}N_3$
was no longer significantly present in the following factors.

**BVOCs-OOMs I factor**

The first nighttime factor has its maximum concentration at around 20:00 LT, and
decreases to very low value during the day. It is moderately correlated with the
production rate of $NO_3$ radical ($P_{NO3}$ derived from Eq. 5) at night, and reaches high
intensity only under conditions of NO below 1 ppb (Fig. 10(a)), indicating a chemical





process of NO₃ radical. The concentration of this factor is mainly from $C_5$ peaks,
followed by $C_6$-$C_{10}$ peaks (Fig. 9(d)), about 80% of which are ONs (Fig. 2(c)),
designating the oxidations of isoprene and monoterpenes by NO₃ (BVOCs-OOMs I).
In the case of isoprene oxidation, the nitrate groups of $C_5H_9O_{4-8}N$, $C_5H_7O_{5-8}N$ and
$C_4H_7O_{5-6}N$ series (summarized in Table S8) are likely to come from the addition of NO₃.
Next, the $C_5H_{10}O_{8-9}N_2$ and $C_5H_8O_{7-10}N_2$ series are probably second-generation products.
These compounds derived from isoprene+NO₃ system have been discussed in previous
laboratory (Kwan et al., 2012;Zhao et al., 2020) and ambient data sets (Ayres et al.,
2015;Xiong et al., 2015). Additionally, The $C_6$-$C_{10}$ species are potentially the products
of monoterpenes degraded by NO₃.

$$P_{NO_3} = k_{NO_2+O_3} \cdot [NO_2] \cdot [O_3] \qquad (5)$$

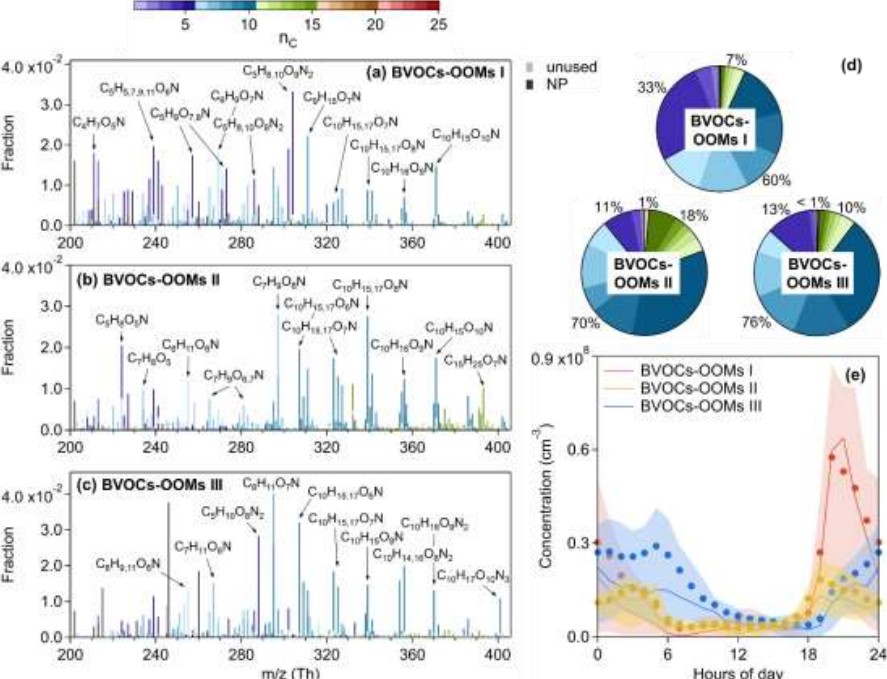

Figure 9. Mass spectra of (a) the BVOCs-OOMs I factor, (b) the BVOCs-OOMs II
factor, (c) the BVOCs-OOMs III factor, and the elemental formulas of major peaks are
labeled above them. Peaks are color-coded by $n_C$ as indicated at the top of the figure,
and the fractions of peaks grouped by $n_C$ are reported in (d) the pie chart. The gray
sticks are fluorinated contaminations, or non-identified compounds. The nitrated
phenols are drawn separately with black peaks in (a), (b) and (c). The molecules
represented by the gray and black sticks were not included in (d). Diurnal patterns of
these three factors are shown in (e), the bold solid lines are the median values, shaded
areas represent percentiles of 75 % and 25 % and solid circles represent mean values.






**BVOCs-OOMs II factor**

The second nighttime factor are intense at night and over five times lower during the
day. Like the BVOCs-OOMs I factor, this factor has high concentrations when NO is
reduced leading to increased $NO_3$ availability (Fig. 10(b)), and about 80% of
compounds in this factor are ONs (Fig. 2 (c)). Accordingly, this may also be a factor
strongly influenced by $NO_3$. It is dominated by $C_6$-$C_{10}$ OOMs, among which the highest
intensity is at $C_{10}$ (Fig. 9(d)), coherent with the nature of monoterpene products
(BVOCs-OOMs II). This factor has weaker signals at $C_{15}$ which are plausibly the
products of sesquiterpenes but could also be dimmers formed from R4 (monoterpenes
+ isoprene or monoterpenes + $C_5$ monoterpenes fragments). Compared to the BVOCs
OOMs I factor (Fig. 9(d)), this factor has more large mass molecules ($C_{10}$) and fewer
small mass molecules ($C_5$), resulting in an effective volatility over one order of
magnitude lower. A $NO_3$-initiated factor, called the nighttime type-2 factor, has also
been discovered in Hyytiälä, Finland (Yan et al., 2016), but the similar factor we found
has a higher proportion of organic nitrates, due to the more abundant $NO_x$ here.

$$RO_2 \bullet \ + \ R'O_2 \bullet \ \rightarrow \ ROOR' \ + \ O_2 \qquad (R4)$$

$$C_{10}H_{16} + NO_3 \bullet \ \xrightarrow{O_2} \ C_{10}H_{16}NO_5 \bullet \ \xrightarrow{H-shift+O_2} \ C_{10}H_{16}NO_7 \bullet$$
$$\xrightarrow{H-shift+O_2} \ C_{10}H_{16}NO_9 \bullet \ \xrightarrow{H-shift+O_2} \ C_{10}H_{16}NO_{11} \bullet$$
(R5a)

$$C_{10}H_{16} + NO_3 \bullet \ \xrightarrow{O_2} C_{10}H_{16}NO_5 \bullet \rightarrow \ C_{10}H_{16}NO_4 \bullet \ \text{(Alkoxy)}$$
$$\xrightarrow{H-shift+O_2} \ C_{10}H_{16}NO_6 \bullet \ \xrightarrow{H-shift+O_2} \ C_{10}H_{16}NO_8 \bullet$$
(R5b)

In terms of fingerprint molecules of this factor (summarized in Table S9), The
$C_{10}H_{15}O_{5-12}N$ series are carbonyl products from precursor $RO_2$ or RO terminations,
while the $C_{10}H_{17}O_{5-9}N$ series are alcohol or hydroperoxide products from precursor $RO_2$
terminations. The $C_7H_9O_{6-8}N$, $C_9H_{15}O_{6-9}N$, $C_9H_{13}O_{7-10}N$, and $C_8H_{13}O_{7-8}N$ series are
expected to be fragments. The closed-shell compounds mentioned above have been
reported in the experiments of monoterpenes + $NO_3$ system (Nah et al., 2016;Faxon et
al., 2018;Takeuchi and Ng, 2019).

It is noteworthy that a set of nitrogen-containing radicals, $C_{10}H_{16}O_{6-11}N$ (peak fitting
are shown in Fig. S8), is present in the BVOCs-OOMs II factor. Starting from a generic
monoterpene molecule with the formula $C_{10}H_{16}$, the $NO_3$ addition with fast $O_2$ addition
results in a peroxy radical with the formula $C_{10}H_{16}O_5N$, If the initial intermediate $RO_2$





is capable to proceed via autoxidation by the formal addition of $O_2$, we expect radicals,
$C_{10}H_{16}O_{5+2x}N$ (x denotes times of autoxidation performed) with an odd oxygen number,
to be formed (R5a). In addition, peroxy radicals with an even oxygen number,
$C_{10}H_{16}O_{6+2x}N$, are likely produced via reaction chain 5b: (1) $RO_2$ is propagated to RO
through bimolecular reactions, and (2) RO isomerize to an alcohol by internal H
abstraction forming a carbon-centered radical (Orlando et al., 2003;Orlando and
Tyndall, 2012), (3) the carbon-centered radical can again take up an oxygen molecule
and follow the autoxidation route. The $C_{10}H_{16}O_9N$ radical is also moderately intense in
the BVOCs-OOMs I factor (Fig. 9(a)), testifying the presence of $NO_3$ chemistry. These
$C_{10}H_{16}O_{6-11}N$ radicals are also reported in the CLOUD chamber (Yan et al., 2020). In
addition to $C_{10}$ radicals, a $C_5$ radical, $C_5H_8O_5N$ (peak fitting are shown in Fig. S8), is
also found in the BVOCs-OOMs II factor. $C_5H_8O_5N$ are possibly derived from the
oxidation of isoprene initiated by $NO_3$, as observed in the laboratory (Zhao et al., 2020).
Another hypothesis is that $C_5H_8O_5N$ is formed from the fragmentation process of
monoterpenes.

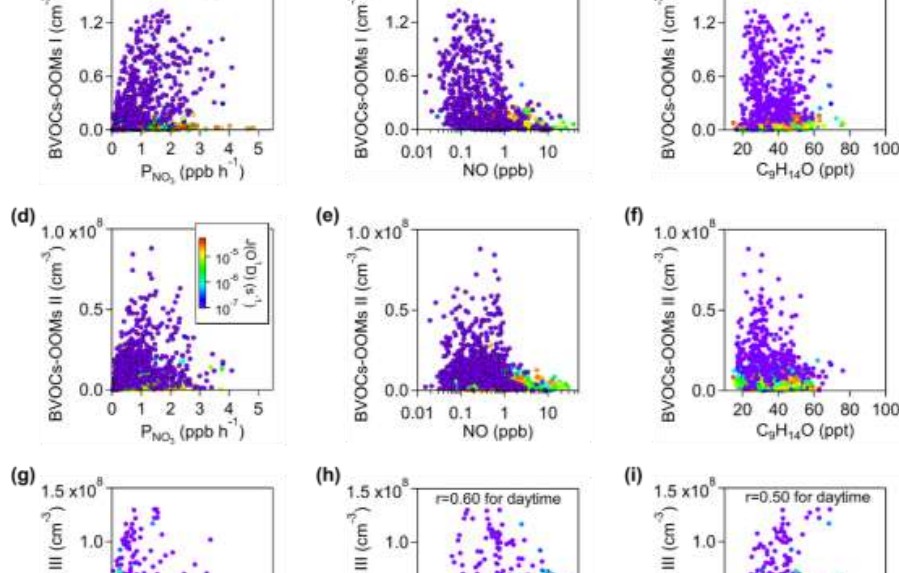


Figure 10. Scatter plots of the BVOCs-OOMs I factor with (a) $P_{NO_3}$, (b) NO, and (c)
nopinone ($C_9H_{14}O$). Scatter plots of the BVOCs-OOMs II factor with (d) $P_{NO_3}$, (e) NO,
and (f) nopinone ($C_9H_{14}O$). Scatter plots of the BVOCs-OOMs III factor with (g) $P_{NO_3}$,





(h) NO, and (i) nopinone ($C_9H_{14}O$). All the scatters are colored by $J(O^1D)$, to show the
difference between day and night. Pearson correlation coefficient showed in (a) is
calculated for nighttime, but the correlation coefficients in (c) are only for daytime.
**BVOCs-OOMs III factor**
The third nighttime factor (BVOCs-OOMs III) is dominated by nitrogen-containing
species with a contribution ratio about 90%, among which dinitrates account for more
than 20% (Fig. 2(c)). When grouped by carbon numbers, $C_{10}$ OOMs have the strongest
signal. Unlike the above two $NO_3$-related factors, this factor shows no correlation with
$P_{NO3}$, but has positive correlation with NO, especially during the daytime (Fig. 10(c)).
$C_9H_{14}O$, a typical product of NO-affected monoterpenes oxidation (Calogirou et al.,
1999), is found to be correlated with this factor (Fig. 10(c)). It is reasonable to infer that
these organic nitrates may come from terminations of monoterpenes-$RO_2$ by NO. In
addition to the elevated intensity during the nighttime, this factor still remains at a
relatively high concentration in the morning, which is much higher than that of the two
$NO_3$-related factors (Figure 9(e)). Owing to the suppression of NO to $RO_2$ autoxidation
and the relatively insufficient oxidant in dark environment, the effective OSc of the
BVOCs-OOMs III factor is lower than other factors. Apart from the mononitrates
summarized in Tabel S10, the $C_{10}H_{16}O_{7-10}N_2$ (dinitrates) and $C_{10}H_{17}O_{10}N_3$ (a trinitrate
charged by $NO_3^-$ at m/z 401) are most likely the result of multiple-generation processes
involving OH or $NO_3$ oxidation of monoterpenes proceeding $RO_2$ + NO terminations.
A similar factor, called terpene nitrates, has also been reported in Centreville, USA
(Massoli et al., 2018), while in Hyytiälä, Finland (Yan et al., 2016), it's that the daytime
type-1 factor is related to NO.
**3.4 Nitrated phenols factors**
Nitrated phenols are of concern, because of their phytotoxicity (Rippen et al., 1987)
and as an important chromophores of brown carbon in aerosol (Desyaterik et al.,
2013;Mohr et al., 2013). The sources of these highly volatile compounds are attributed
to biomass burning, vehicle exhausts, and secondary gas or aqueous phase production
(Harrison et al., 2005). Here we identified three factors about NP, including the NP I
factor dominated by nitrophenol, the NP II factor dominated by substituted nitrophenols,
and the NP III factor dominated by dinitrophenols. Although the mass spectrum of the
NP III factor is less pure than the NP I & II factors (Fig. 11), its time series follows well
with $C_6H_4O_5N_2$ (Fig. 11(f)), implying that this factor is driven by di-nitrated-phenols
chemistry. Since nitrated phenols have been broadly investigated and relatively clearly
recognized (Harrison et al., 2005;Yuan et al., 2016;Wang et al., 2018b;Cheng et al.,
2021), they are not discussed too much here. It seems that the chemistry of nitrated
phenols is distinctive to other OOMs.



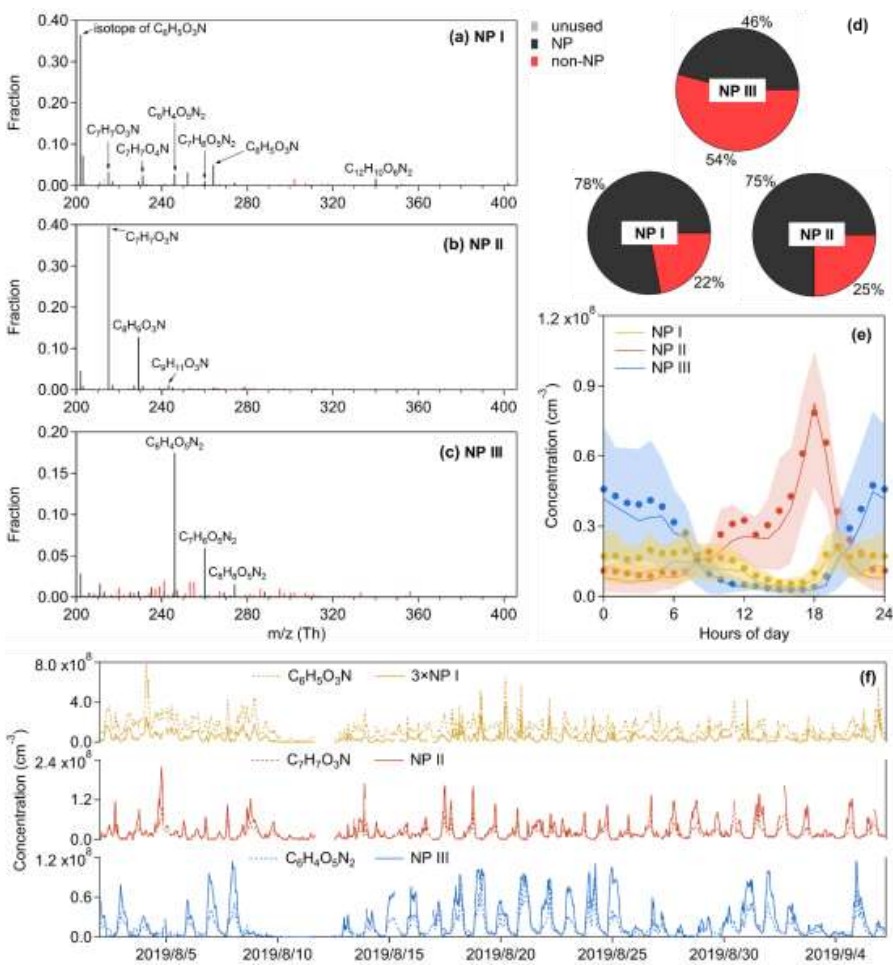


Figure 11. Mass spectra of (a) the NP I factor, (b) the NP II factor, and (c) the NP III
factor, and the elemental formulas of major peaks are labeled above them. The gray
sticks are fluorinated contaminations, or non-identified compounds. The nitrated
phenols are drawn separately with black peaks in (a), (b) and (c), while other OOMs
are plotted as red peaks. The molecules represented by the gray were not included in
(d). Diurnal patterns of these three factors are shown in (e), the bold solid lines are the
median values, shaded areas represent percentiles of 75 % and 25 % and solid circles
represent mean values. (f) Time series of PMF factors and tracers.

**3.5 Ensemble chemical properties**

After performing PMF analysis, over 1000 non-nitro molecules have been identified
through HR peaks fitting in each factor. The mean concentration of total non-nitro
OOMs reconstructed from the selected PMF solution is about $2.1 \times 10^8$ molecules cm$^{-3}$.
Ensemble chemical properties of these non-nitro OOMs are summarized in Fig. 12. The





number of carbon atoms implies the precursor information of OOMs. $C_5$ OOMs, which
principally consist of isoprene products benefited from the high reactivity and intensive
emissions of isoprene in summer, are the most abundant (Fig. 12(c)). While $C_6$-$C_9$
OOMs are mostly likely formed from the oxidation of AVOCs such as aromatics and
aliphatic series in the urban and suburban atmosphere, and as we expected, these
AVOCs-derived OOMs account for about 50% of the total signal (Fig. 12(c)). The
intensity of OOMs decreases from $C_7$ to $C_9$ determined by the concentration distribution
of precursors, but becomes a plateau at $C_{10}$ (Fig. 12(c)), indicating another source of
$C_{10}$ OOMs, such as monoterpenes oxidation. These results underscore the formation of
SOA precursors from a mixture of anthropogenic and biogenic emissions, under
ongoing forest cover increases (Wang et al., 2020a) in highly urbanized eastern China.
In addition to the anthropogenic VOCs, another human-induced perturbation on the
formation of OOMs is the $NO_x$-affected chemistry of VOCs, i.e., $RO_2$ + NO
terminations or $NO_3$-iniated oxidations. As showed in Fig. 12(c), about 70% of OOMs
are nitrogen-bearing compounds, regarded as organic nitrates within the allowed range
of uncertainty. If isoprene nitrates are not included, organic nitrates peak at $C_7$ as do the
nitrogen-free species, showing the significant production of organic nitrates through the
AVOCs + $NO_x$ pathways. The $NO_x$ effect on AVOCs-derived OOMs, typified by the
Aro-OOMs factor and the Aliph-OOMs factor, are not showed in previous ambient
measurements (Yan et al., 2016;Lee et al., 2016;Massoli et al., 2018).
OOMs grouped by carbon numbers or nitrogen numbers consistently have absolute
high concentrations in the daytime (Fig. 12(a) and (b)), revealing the crucial role of
photochemical progress, involving $RO_2$ + NO termination reactions, on OOMs. In
addition, The $C_5$-$C_{10}$ OOMs are enhanced again during 19:00-22:00 LT, and the
nighttime peak of $C_{10}$ OOMs is even higher than its daytime peak (Fig. 12(a)). The
nocturnal $C_{10}$ OOMs are more intense than $C_9$ OOMs (Fig. 12(a)), and there are more
$C_{10}$ nitrates than $C_9$ nitrates (Fig. 12(c)). These results show the fate of VOCs degraded
by $NO_3$ during the nighttime, which are more important to monoterpenes. In contrast to
nitrogen-free OOMs, organic nitrates are enriched through the reactions of BVOCs with
$NO_3$ in the early evening (Fig. 12(b)), as indicated by three BVOCs nighttime chemistry
factors.
Apart from reflecting the influence of $NO_x$, multi-nitrates also imply the multiple
generations of VOCs oxidation, which is evident in the products of isoprene (e.g.,
$C_5H_{10}O_8N_2$ and $C_5H_9O_{10}N_3$) due to its two carbon-carbon double bonds. As products of
mononitrates, multi-nitrates follow mononitrates in diurnal variation, with double peaks
initiated by OH and $NO_3$ respectively (Fig. 12(b)). Considering that the formation of
organic nitrate is only a small branch of $RO_2$ + NO termination, the contribution of
multi-step oxidation should be larger than that shown in Fig. 12(c).



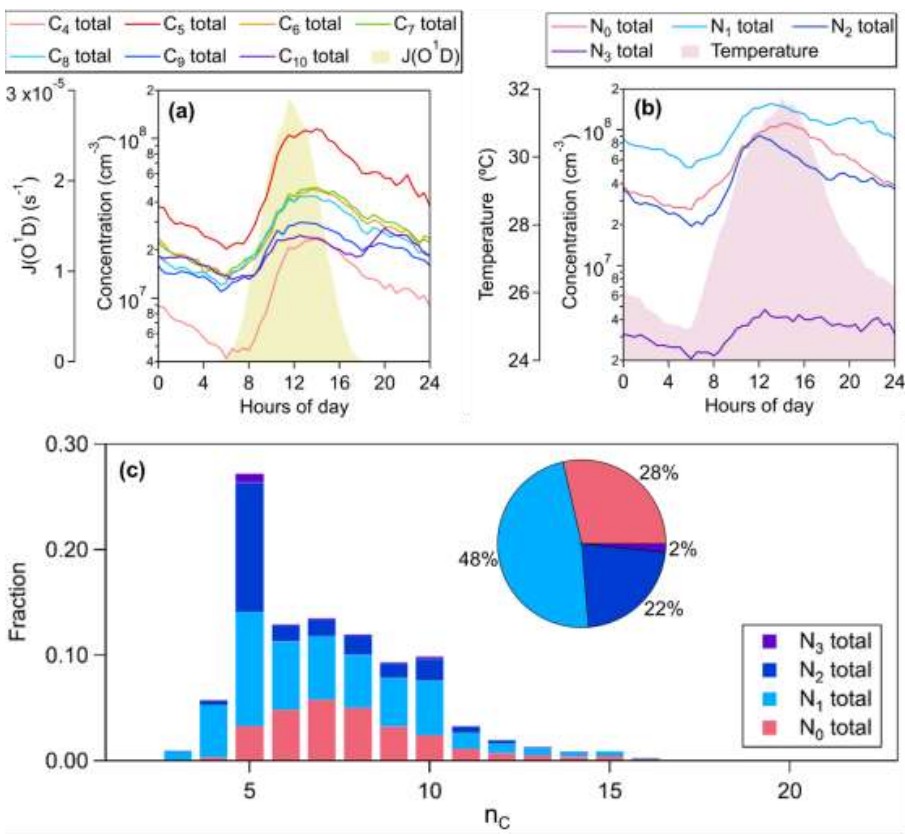

Figure 12. Ensemble chemical properties of non-nitro OOMs reconstructed from the selected PMF solution. (a) Median diurnal cycles of total compounds with carbon number of 5-10 respectively. (b) Median diurnal cycles of total compounds with $n_N$ of 0-3 respectively. (c) The distributions of total observed OOMs at different $n_C$. OOMs on each carbon number is grouped by nitrogen number, and the total concentration fractions of each groups are reported in the pie chart. Since we selected peaks in the m/z range of 202-404 Th, OOMs with $n_C < 5$ or $n_C > 10$ detected by nitrate CI-APi-TOF are underestimated.

**4 Conclusions**

We have investigated the sources and characteristics of gas-phase OOMs observed using a nitrate CI-APi-TOF at the SORPES station in the YRD of eastern China, an environment dominated by anthropogenic emissions with enhanced biogenic emissions during summer.

The binPMF analysis, which avoids the uncertainty introduced by high-resolution peak fitting to the input data matrix, was applied to deconvolve the complexity of the data set, and it resolved 14 factors, among which 12 factors have been discussed in detail. A



morning factor (Aro-OOMs), correlated with the production rates of $RO_2$ from
aromatics, is characterized by unsaturated products of aromatics such as $C_xH_{2x-5}O_{6-9}N$
(x=6-12). An afternoon factor (Aliph-OOMs), containing the bulk of $C_6$-$C_9$ dinitrates
and trinitrates such as $C_xH_{2x-2}O_8N_2$ (x=4-13) and $C_xH_{2x}O_8N_2$ (x=4-8), is assumed to be
derived from aliphatics oxidation. A transported factor (Isop-OOMs), correlates with
MVK / MACR and SOA, is exclusively dominated by isoprene nitrates (e.g.,
$C_5H_{10}O_8N_2$ and $C_5H_9O_{10}N_3$). A nighttime factor (BVOCs-OOMs III), related to NO, is
dominated by terpenes nitrates such as $C_{10}H_{15}O_6N$, $C_{10}H_{16}O_{7-10}N_2$ and $C_{10}H_{17}O_{10}N_3$. In
addition to the factors distinguished by precursors, several factors are driven by
chemistry. A factor following the $J(O^1D)$ (Photo-related), consisting of isoprene
products mixed with others, is thought to be produced by in situ photochemistry. An
afternoon factor (Temp-related), having the most abundant nitrogen-free OOMs such
as $C_xH_{2x-4}O_{5-6}$ (x=5-10), $C_xH_{2x-2}O_5$ (x=5-10), and $C_xH_{2x-6}O_4$ (x=5-10), is generated
involving temperature-influenced chemistry. A daytime factor ($O_x$ & SOA-related),
correlated well with $O_x$ and SOA, indicates the photochemical aging process. Two
nighttime factors (BVOCs-OOMs I & II), benefiting from $NO_3$ and suppressed by NO,
are considered to be produced from $NO_3$-iniated oxidation of BVOCs, and both of them
have the fingerprint molecule, $C_{10}H_{16}O_9N$. The remaining three factors are governed
by nitrated phenols.

All of these factors from various precursors are influenced in different ways by $NO_x$.
Over 1000 non-nitro molecules have been identified and then reconstructed from
selected solution of binPMF, and about 72% of the total signal are contributed by
nitrogen-containing OOMs, almost regarded as organic nitrates formed through $RO_2$ +
NO terminations or $NO_3$-initiated oxidations. Moreover, multi-nitrates have a
contribution ratio of about 23% to total concentration, indicating the significant
presence of multiple oxidation generations, especially for isoprene (e.g., $C_5H_{10}O_8N_2$
and $C_5H_9O_{10}N_3$). The nitrate CI-APi-TOF data set presented here highlight the decisive
role of $NO_x$ chemistry on OOMs formation in densely populated areas. In summary,
our findings highlight the dramatic interactions between anthropogenic and biogenic
emissions, and encourage more investigations from a mechanistic point of view.

The differences in OOMs observed in different environments are clear, and the
underlying causes are well worth considering. The precursors, oxidants, and formation
pathways of OOMs change when moving from urbanized areas to pristine regions, as
AVOCs and NOx concentrations decrease and BVOCs concentrations increase. This
process can also occur under the trend of global warming and anthropogenic emissions
mitigation, but we still know very little about it. Clarifying the variations of
composition, properties, and formation efficiency of OOMs will help to understand the
evolution of SOA production during this process.

**Appendix A. The selected solution of binPMF analysis on nitrate CI-APi-TOF data**

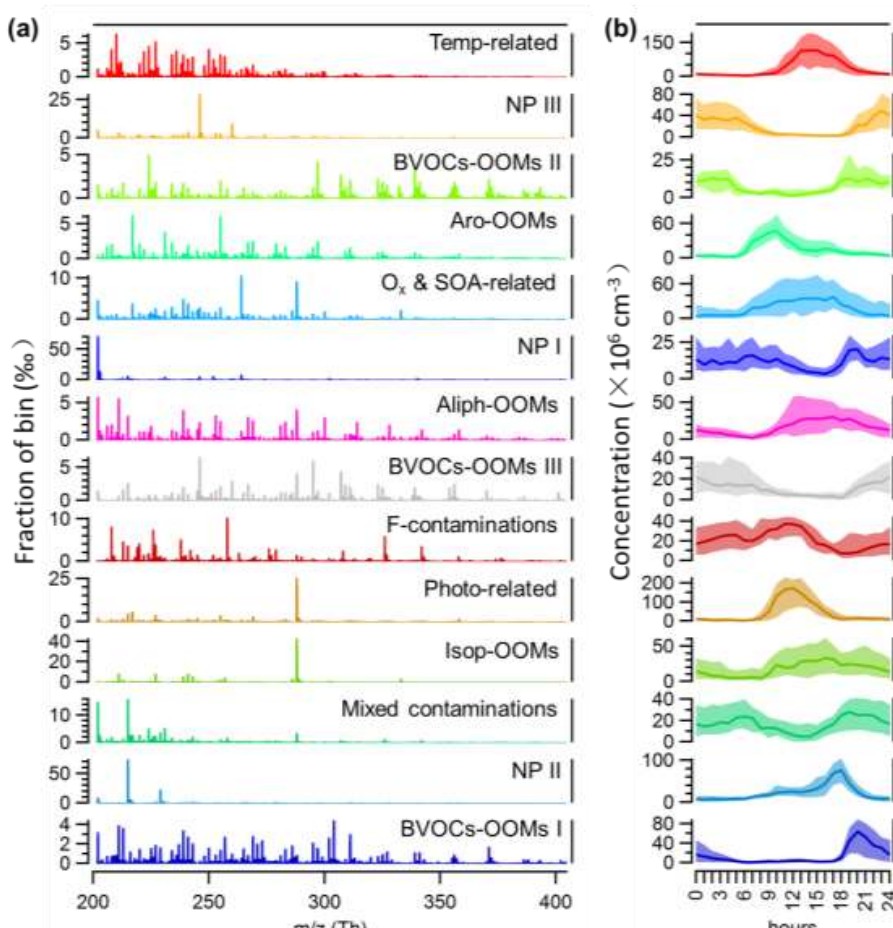


Figure A1. The selected solution for binPMF analysis of nitrate CI-APi-TOF data,
showing (a) mass profile and (b) diurnal cycle of different factor.

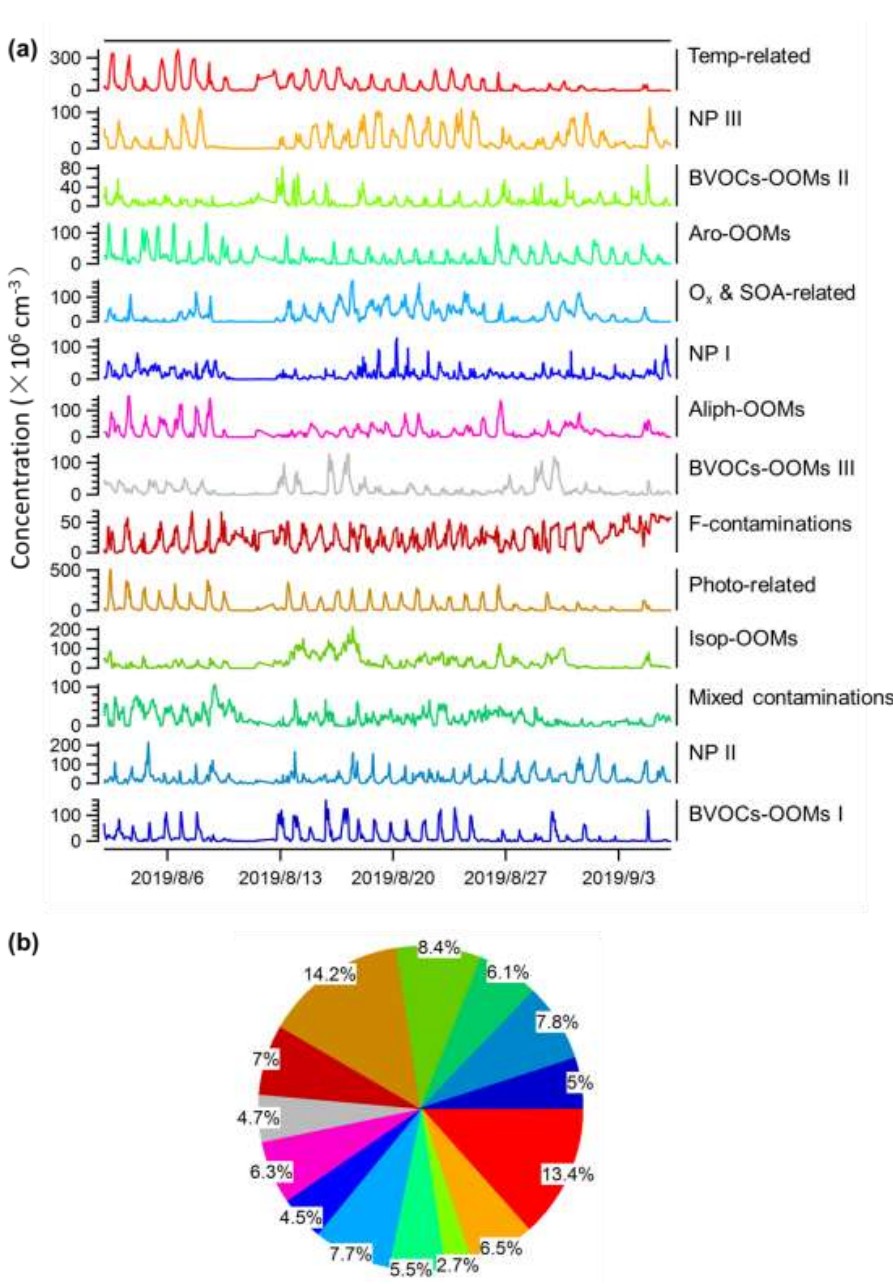


Figure A2. The selected solution for binPMF analysis of nitrate CI-APi-TOF, showing
(a) time series of and (b) contribution to total signal reconstructed by PMF of each
factor.

**Data availability.** Measurement data at the SORPES station, including OOMs data and
relevant trace gases and aerosol data as well as meteorological data, are available upon





request from the corresponding author before the SORPES database is open to the
public.
**Author contributions.**

**Competing interests.** The authors declare that they have no conflict of interest.
**Acknowledgements.** We thank colleagues and students at the School of Atmospheric
Sciences at Nanjing University for their contributions to the maintenance of the
measurements. We thank the tofTools team for providing tools for mass spectrometry
analysis**.**
**Financial support.** This work was mainly funded by the National Key R&D Program
of China (2016YFC0202000 and 2016YFC0200500), the National Natural Science
Foundation of China (NSFC) project (41875175, 42075101 and 92044301),
the Shanghai Rising-Star Program (19QB1402900), and Jiangsu Province Key R&D
Program Major Technology Demonstration (BE 2019704).

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
