# Peer review of "Formation of condensable organic vapors from anthropogenic and"

_Atmospheric Chemistry and Physics, 2021_

## Author Comment (AC1)

This manuscript utilized an improved source apportionment approach that has been developed recently, binPMF, to deconvolve nitrate CI-APi-TOF mass spectra from a highly-developed, densely-populated urban region in eastern China. The authors identified factors from different sources and discussed the influence of anthropogenic and biogenic emissions. Overall, the manuscript is well written and scientifically interesting. I recommend publication after the following comments are addressed.

1. All figures have a poor resolution. Formatting issues? Please update.
Response: Thanks for the comment. We have updated all figures in the revised manuscript.

2. Equation 1: Can the authors provide more details about calibration factor determination? Are the inlet configuration and flow rate in the reference the same as these used in this work? Moreover, the reagent ion can have different sensitivities towards different compounds, and the sensitivities also vary for ions from the same species but charged in different ways. Can the authors validate the use of $H_2SO_4$-based calibration factor for all species and elaborate more about the potential impact on the results?
Response: Thanks for the suggestion.

The deployment of mass spectrometry in atmospheric observations has allowed us to measure thousands of molecules, but has also brought great challenges to instrument calibration. The unknown and diverse molecular structures make it impossible to provide all standard molecules to calibrate and quantify all the molecules. Therefore, we used an empirical method for quantifying OOMs based on the ionization kinetics (pseudo first order reaction approximation) in the reaction tube of the chemical ionization source (Eq. 1) (Heinritzi et al., 2016).

First, we calibrated the sulfuric acid according to the method proposed by Kuerten et al. (2012) during the observation period. Briefly, providing a known concentration of gaseous sulfuric acid by connecting the nitrate CI-APi-TOF to a well-characterized $H_2SO_4$ generator, and from comparisons of multiple $H_2SO_4$ concentrations and the signals measured by the nitrate CI-APi-TOF, a value of 4.2e9 molecules $cm^{-3}$ for the $H_2SO4$-based calibration factor C, proceeding taking into account the diffusion loss in the sampling line, was obtained. The reaction where nitrate clusters react with $H_2SO_4$ has proceeded at the collision limit (Viggiano et al., 1997). The uncertainty of C obtained by this method is 33% (Kuerten et al., 2012).

Standards for oxygenated organic molecules (OOMs) measurable by the nitrate CI-APi-TOF are still lacking. Quantum chemistry computations showed that at least two hydrogen bond donor functional groups are needed for an oxygenated organic molecule

to be detected in a nitrate CI-APi-TOF (Hyttinen et al., 2015), and when the number of hydrogen bond donating functional groups in the target molecule is greater than or equal to 2, the binding of the target molecule to the reagent ion depends almost linearly on the number of oxygen atoms in the target molecule (Hyttinen et al., 2018). Almost all OOMs analyzed in our study have oxygen numbers greater than or equal to 4, which fits the above law. Ehn et al. (2014) employed several methods, both empirical and theoretical, to estimate the sensitivity of the nitrate CI-APi-TOF to highly oxygenated organic molecules (HOMs, with oxygen numbers greater than or equal to 6). They found that the collision frequencies of $(HNO_3)_x(NO_3^-)$, $x = 0\text{-}1$, with HOMs were comparable to those of nitrate clusters with $H_2SO_4$, which means that the sensitivities of nitrate clusters to $H_2SO_4$ and HOMs can be assumed equal. Finally, they estimated a ±50% uncertainty in reported concentrations of HOMs.

Although one can speculate the assumption that the instrument acquires all molecules with the same ionization efficiency may suffer uncertainties, the calibration procedure for OOMs ($H_2SO_4$ based calibration factor + mass-dependent transmission efficiency), as a relatively accurate method, is so far the optimized method and has been adopted in many studies (Kirkby et al., 2016a;Trostl et al., 2016;Stolzenburg et al., 2018). And to our best knowledge, there is still no way to calibrate the nitrate chemical ionization source for charging efficiency for a wide range of molecules. The sensitivity of the instrument for some moderately oxygenated organic molecules, not HOMs, are weaker than that of $H_2SO_4$. Although we cannot provide an exact uncertainty of these OOMs concentration, some implications can be obtained from previous studies. Ehn et al. (2014) found that the calibration factor of perfluoroheptanoic acid was 3-4 times higher than that of $H_2SO_4$ in the nitrate CI-APi-TOF, and Massoli et al. (2018) found that the calibration factor of malonic acid was about 4 times higher than that of $H_2SO_4$. Similarly, the concentrations of moderately oxygenated organic molecules were presumably underestimated by a factor of 4 to match the observation in the study of Trostl et al. (2016). More efforts are needed to accurately quantify these OOMs.

In summary, our calibrations lead towards a lower limit estimate of OOMs concentrations, but it is the optimized option available to our best knowledge. It should be noted that the accurate quantification of OOMs is not the main result of this study and that errors in the quantification of OOMs do not affect our conclusions.

We've rephrased the sentences about calibration in the manuscript for clarity, **line 158-181**:

*Due to the diversity and unknown molecular structures of oxygenated organic compounds, standards for OOMs measurable by the nitrate CI-APi-TOF are still lacking. Like other studies have done (Kirkby et al., 2016b;Trostl et al., 2016;Stolzenburg et al., 2018), an empirical method was used to quantify the*

*concentrations of OOMs based on the ionization kinetics (pseudo first order reaction approximation) in the reaction tube of CI (Eq. 1) (Heinritzi et al., 2016).*

$$[OOM_i] = ln\left(1 + \frac{\sum_{n=0}^{1}[OOM_i \cdot (HNO_3)_n \cdot NO_3^- + (OOM_i - H)^-]}{\sum_{n=0}^{2}[(HNO_3)_n \cdot NO_3^-]}\right) \times C \times T_i \qquad (1)$$

*Here [$OOM_i$] is the concentration (molecules $cm^{-3}$) of one OOM. On the right side of the equation, the numerator in the parenthesis is the detected total signals (ions/s) of one OOM charged by nitrate ions in forming-adduct or deprotonated ways, the denominator is the sum of all reagent ion signals (ions/s). First, a $H_2SO_4$-based calibration factor C, with a value of $4.2 \times 10^9$ molecules $cm^{-3}$, was obtained from a calibration using $H_2SO_4$ (Kuerten et al., 2012) proceeding taking into account the diffusion loss in the sampling line by assuming that all detected OOMs have the same ionization efficiency as $H_2SO_4$. The collision frequency of HOMs with nitrate clusters is comparable to that of sulfuric acid with nitrate clusters (Ehn et al., 2014;Hyttinen et al., 2015), yet the collision frequency of some moderately oxygenated molecules with nitrate clusters is relatively slower. Therefore, calibration by this method leads to a lower limit estimate of OOMs concentrations (Ehn et al., 2014;Trostl et al., 2016), but the accurate quantification of OOMs is not the main concern of this study and the errors in the quantification of OOMs do not change our conclusions.*

3. Equation (2): Can the authors include more details of the equation, i.e. how CS is calculated, $k_{OH+SO2}$ value (or calculation) and source?

Response: Thanks for the suggestion. We've add these details in the revised manuscript, **line 211-223**:

$$[OH] = \frac{[H_2SO_4] \cdot CS}{k_{OH+SO_2} \cdot [SO_2]} \qquad (2)$$

*Where $k_{OH+SO_2}$ is a termolecular reaction constant for the rate-limiting step of the formation pathway of $H_2SO_4$ in the atmosphere (Finlayson-Pitts and Pitts, 2000), and condensation sink (CS) is the loss rate of $H_2SO_4$ by condensation to aerosol surface. The value of $k_{OH+SO_2}$ is inferred from the IUPAC Task Group on Atmospheric Chemical Kinetic Data Evaluation (https://iupac-aeris.ipsl.fr/, last access: 09 August 2021). The value of CS was calculated following Eq. (3) (Kulmala et al., 2012):*

$$CS = 2\pi D \sum_i \beta_{m_i} d_{p_i} N_i \qquad (3)$$

*Where D is the diffusion coefficient of gaseous sulfuric acid, $\beta_m$ is a transition-regime correction factor dependent on the Knudsen number (Fuchs and Sutugin, 1971), and $d_{p_i}$ and $N_i$ are the diameter and number concentration of particles in size bin i.*

4. Line 212: "the raw spectra with were…" Some words seemed missing here.
Response: We've re-phased these sentences in the revised manuscript, **line 239**:

*Briefly, the raw spectra were divided into narrow bins with a width of 0.006 Th after mass calibration.*

5. Figure 1: Can the authors add the diurnal patterns of all parameters? The NO, temperature, and J(O¹D) diurnals were included in Figure 4 and 12, but it would be good to have a summary plot.
Response: Thanks for the suggestion. We've add the suggested plot in the supplement (Fig. S1).

[Figure]

*Fig. S1. Median diurnal variations of (a) J(O¹D), Temperature, NO, total aromatics (benzene + toluene + C8 aromatics + C9 aromatics + C10 aromatics + styrene), and isoprene, and (b) mass spectra of nitrate CI-APi-TOF with m/z in the range of 201-404 Th.*

6. Table 1: As mentioned before, the authors may need to consider the effects of

assuming a constant ionization efficiency. How credible are the reported concentrations? Can the authors include uncertainties?

Response: As the details provided in the response of second comment, HOMs can be well measured by the nitrate CI-APi-TOF with a ±50% uncertainty in quantification (Ehn et al., 2014). For other OOMs (moderately oxygenated organic molecules), an exact uncertainty is not available. We inferred from previous studies that these OOMs may be underestimated by a factor of 3-4 (Ehn et al., 2014;Massoli et al., 2018;Trostl et al., 2016). By the way, we did not intend to emphasize the quantitative concentrations of OOMs in this study. We've rephrased the statement as follows in the revised manuscript, **line 175-181**:

*The collision frequency of HOMs with nitrate clusters is comparable to that of sulfuric acid with nitrate clusters (Ehn et al., 2014;Hyttinen et al., 2015), yet the collision frequency of some moderately oxygenated molecules with nitrate clusters is relatively slower. Therefore, calibration by this method leads to a lower limit estimate of OOMs concentrations (Ehn et al., 2014;Trostl et al., 2016), but the accurate quantification of OOMs is not the main concern of this study and the errors in the quantification of OOMs do not change our conclusions.*

7. Line 341: The authors used "autoxidation" instead of "auto-oxidation" elsewhere. Please be consistent. Moreover, the diurnal pattern of Aro-OOMs almost followed that of NO, would autoxidation be suppressed?

Response: Thanks for the comment and suggestion.

First, we have uniformly used the word "autoxidation".

Second, NO does suppress autoxidation, but the reactions are supposed to be very complex. As showed in Fig. 4(f), the peak time of Aro-OOMs is later than that of NO, but overlaps with the peak time of the production rates of OH-initiated primary $RO_2$ from aromatics. Hence, we suggest that Aro-OOMs are mainly controlled by the photo-oxidation of aromatics, i.e., the source of $RO_2$, while the influence from the sink of $RO_2$ may be minor, e.g., $RO_2$ +NO.

8. Figure 8: One obvious difference between the two isoprene oxidation products from this figure is that $C_5H_{10}O_8N_2$ was mostly attributed to Isop-OOMs and Photo-related factors, while $C_5H_9O_{10}N_3$ was to Isop-OOMs and $O_x$ & SOA-related factors. As the authors proposed that $C_5H_9O_{10}N_3$ was more likely to be transported than $C_5H_{10}O_8N_2$, does this imply that the $O_x$ & SOA-related factor was transported?

Response: Thanks for the comment and suggestion. From the molecular formulas, we can assume that $C_5H_9O_{10}N_3$ is a bit more 'aged' than $C_5H_{10}O_8N_2$, but the chemical aging

processes and regional transport are not directly correlated. An air mass that stays local can also keep aging. We prefer saying that during the period of August 13 to August 17, 2019 (Fig. 8), $C_5H_9O_{10}N_3$ was almost entirely transported (**mainly from the Isop-OOMs factor**), while $C_5H_{10}O_8N_2$ has strong in situ photochemical generation (from the Photo-related factor), in addition to the source of transport (from the Isop-OOMs factor). As for the $O_x$ & SOA-related factor, its time series (Fig. A2) seems to be composed of background concentrations (having episodes) and photochemical production (having diurnal variation). We speculate that this factor comes from the photo-oxidation process of VOCs, which produces ozone and SOA concurrently and may occur regionally. We need more observations and analysis to confirm the contribution of regional transport to the observed Isop-OOMs.

**Reference**

[revised manuscript text omitted]

---

## Author Comment (AC2)

This paper discussed the measurements of oxygenated organic molecules (OOMs) performed in Nanjing in eastern China. A nitrate-ion-based chemical ionization mass spectrometer was used to perform these ambient measurements, and source apportionment analysis was performed using a recently developed approach positive matrix factorization on binned mass spectra (binPMF). The authors reported several factors related to anthropogenic VOCs daytime chemistry and biogenic VOCs (BVOCs) chemistry, and they discussed the influences of anthropogenic and biogenic emissions on the formation and evolution of these factors. In general, the manuscript was very well-written and the results were presented in a very clear, coherent manner. The topic is of interest to the atmospheric community. I recommend publication after the authors have addressed the following comments:

1.  More details need to be provided for the quantification of OOMs (equation 1) especially since OOMs are the focus of this paper. How was equation 1 derived?

Response: Thanks for the comment.

Equation 1 was derived based on the theoretical framework for deriving concentrations of a certain compound from ion count rates for the CI-APi-TOF, and has been described in detail in the study of Heinritzi et al. (2016):

*Ionization of a compound X via primary ions $P^{\pm}$ can be described by the following reaction:*

$$P^{\pm} + X \xrightarrow{k} X^{\pm} + P$$

*The relation between primary ion concentration $[P^{\pm}]$ and the concentration of the desired compound $[X]$ in the reaction tube of the instrument is given by the following expression, where k is the reaction rate constant for the ionization reaction:*

$$\frac{d[P^{\pm}]}{dt} = -k[P^{\pm}][X]$$

*Assuming [X] as constant over the reaction time t (pseudo first order reaction approximation) enables a simple integration. The additional assumption that the total amount of charge is constant in the reaction tube leads to the following expression for the concentration of [X]:*

$$[X] = \frac{1}{kt} \ ln(1 + \frac{[X^{\pm}]}{[P^{\pm}]})$$

*The concentration of compound X is proportional to the ratio of concentrations of product ions $[X^{\pm}]$ and primary ions $[P^{\pm}]$. However, the mass spectrometer does not*

*measure the ion concentrations in the reaction chamber directly, but only ion count rates at the location of the detector of the instrument. Therefore replacing the concentrations $[X^{\pm}]$ and $[P^{\pm}]$ by ion count rates $i(X^{\pm})$ and $i(P^{\pm})$ is only justified if we account for the relative mass discrimination of the two ion species inside the mass spectrometer:*

$$[X] = \frac{1}{kt} \ ln(1 + \ \frac{i(X^{\pm})}{i(P^{\pm})}) \cdot \frac{1}{T_X}$$

*Here, $T_X$ is the factor that describes the mass discrimination of ions $X^{\pm}$ relative to that of the primary ions $P^{\pm}$. This factor can be strongly dependent on the m/z ratio of the involved ions.*

The reaction rate constants (k) for the ionization of each compound X via primary ions is hard to get. We obtain an apparent factor for $\frac{1}{kt}$ by calibrating, i.e., C.

Since this is a common quantification method for the CI-APi-TOF (Kirkby et al., 2016b; Stolzenburg et al., 2018; Trostl et al., 2016), we only summarize the main points of Eq. (1) in the manuscript and give references for the reader's further inquiry, **line 160-183**:

*Like other studies have done (Kirkby et al., 2016b; Stolzenburg et al., 2018; Trostl et al., 2016), an empirical method was used to quantify the concentrations of OOMs based on the ionization kinetics (pseudo first order reaction approximation) in the reaction tube of CI (Eq. 1) (Heinritzi et al., 2016).*

$$[OOM_i] = ln \left(1 + \frac{\sum_{n=0}^{1}[OOM_i \cdot (HNO_3)_n \cdot NO_3^- + (OOM_i - H)^-]}{\sum_{n=0}^{2}[(HNO_3)_n \cdot NO_3^-]}\right) \times C \times T_i \qquad (1)$$

*Here $[OOM_i]$ is the concentration (molecules cm$^{-3}$) of one OOM. On the right side of the equation, the numerator in the parenthesis is the detected total signals (ions/s) of one OOM charged by nitrate ions in forming-adduct or deprotonated ways, the denominator is the sum of all reagent ion signals (ions/s). First, a $H_2SO_4$-based calibration factor C, with a value of $4.2 \times 10^9$ molecules cm$^{-3}$, was obtained from a calibration using $H_2SO_4$ (Kuerten et al., 2012) proceeding taking into account the diffusion loss in the sampling line by assuming that all detected OOMs have the same ionization efficiency as $H_2SO_4$. The collision frequency of HOMs with nitrate clusters is comparable to that of sulfuric acid with nitrate clusters (Ehn et al., 2014; Hyttinen et al., 2015), yet the collision frequency of some moderately oxygenated molecules with nitrate clusters is relatively slower. Therefore, calibration by this method leads to a lower limit estimate of OOMs concentrations (Ehn et al., 2014; Trostl et al., 2016), but the accurate quantification of OOMs is not the main concern of this study and the errors in the quantification of OOMs do not change our conclusions. Second, a mass*

2. Why did the authors assume that the detected OOMs have the same ionization efficiencies as sulfuric acid? Have they tested their assumption? What are the uncertainties that arise from using this assumption?

Response: Thanks for the comment.

The reaction of nitrate clusters with $H_2SO_4$ has been found to proceed at the collision limit (Viggiano et al., 1997).

Standards for oxygenated organic molecules (OOMs) measurable by the nitrate CI-APi-TOF are still lacking. Quantum chemistry computations showed that at least two hydrogen bond donor functional groups are needed for an oxygenated organic molecule to be detected in a nitrate CI-APi-TOF (Hyttinen et al., 2015), and when the number of hydrogen bond donating functional groups in the target molecule is greater than or equal to 2, the binding of the target molecule to the reagent ion depends almost linearly on the number of oxygen atoms in the target molecule (Hyttinen et al., 2018). Almost all OOMs analyzed in our study have oxygen numbers greater than or equal to 4, which fits the above law.

Ehn et al. (2014) employed several methods, both empirical and theoretical, to estimate the sensitivity of the nitrate CI-APi-TOF to **highly** oxygenated organic molecules (HOMs, with oxygen numbers greater than or equal to 6). They found that the collision frequencies of $(HNO_3)x(NO_3^-)$, x = 0-1, with HOMs were comparable to those of nitrate clusters with $H_2SO_4$, which means that the sensitivities of nitrate clusters to $H_2SO_4$ and HOMs can be assumed equal. Finally, they estimated a ±50% uncertainty in reported concentrations of HOMs.

The sensitivity of the instrument for some **moderately** oxygenated organic molecules, not HOMs, are weaker than that of $H_2SO_4$. Although we cannot give an uncertainty of these OOMs concentration, some implications can be obtained from previous studies. Ehn et al. (2014) found that the calibration factor of perfluoroheptanoic acid was 3-4 times higher than that of $H_2SO_4$ in the nitrate CI-APi-TOF, and Massoli et al. (2018) found that the calibration factor of malonic acid was about 4 times higher than that of $H_2SO_4$. Similarly, the concentrations of moderately oxygenated organic molecules was presumably underestimated by a factor of 4 to match the observation in the study of Trostl et al. (2016).

Although one can speculate the assumption that the instrument acquires all molecules

with the same ionization efficiency may suffer uncertainties, the calibration procedure for OOMs ($H_2SO_4$ based calibration factor + mass-dependent transmission efficiency), as a relatively accurate method, is so far the optimized method and has been adopted in many studies (Kirkby et al., 2016a; Stolzenburg et al., 2018; Trostl et al., 2016). And to our best knowledge, there is still no way to calibrate the nitrate chemical ionization source for charging efficiency for a wide range of molecules. It should be noted that the accurate quantification of OOMs is not the main result of this study and that errors in the quantification of OOMs do not affect our conclusions.

We've stated this in the revised manuscript, **line 175-181**:

*The collision frequency of HOMs with nitrate clusters is comparable to that of sulfuric acid with nitrate clusters (Ehn et al., 2014; Hyttinen et al., 2015), yet the collision frequency of some moderately oxygenated molecules with nitrate clusters is relatively slower. Therefore, calibration by this method leads to a lower limit estimate of OOMs concentrations (Ehn et al., 2014; Trostl et al., 2016), but the accurate quantification of OOMs is not the main concern of this study and the errors in the quantification of OOMs do not change our conclusions.*

3.  The authors stated that VOCs were measured using a PTR-MS. Was the data used in this paper? It was not clear to me whether and/or how the data was used to support results discussed in this paper.
Response: Thanks for the comment. We calculated the rates of $RO_2$ production from some VOCs (Fig. 4), and correlated the VOCs and OOMs data to explore the characterization process of OOMs (Fig. 5).

4.  More details need to be provided for equation 2. How was CS calculated?
Response: Thanks for the comment, we've added these details in the revised manuscript, **line 211-223**:

$$[OH] = \frac{[H_2SO_4] \cdot CS}{k_{OH+SO_2} \cdot [SO_2]} \qquad (2)$$

*Where $k_{OH+SO_2}$ is a termolecular reaction constant for the rate-limiting step of the formation pathway of $H_2SO_4$ in the atmosphere (Finlayson-Pitts and Pitts, 2000), and Condensation sink (CS) is the loss rate of $H_2SO_4$ by condensation to aerosol surface. The value of $k_{OH+SO_2}$ is inferred from the IUPAC Task Group on Atmospheric Chemical Kinetic Data Evaluation (https://iupac-aeris.ipsl.fr/, last access: 09 August 2021). The value of CS was calculated following Eq. (3) (Kulmala et al., 2012):*

$$CS = 2\pi D \sum_i \beta_{m_i} d_{p_i} N_i \qquad (3)$$

*Where D is the diffusion coefficient of gaseous sulfuric acid, $\beta_m$ is a transition-regime correction factor dependent on the Knudsen number (Fuchs and Sutugin, 1971), and $d_{p_i}$ and $N_i$ are the diameter and number concentration of particles in size bin i.*

5.   Were calibrations performed during the field campaign? If no, how confident are the authors that the ionization efficiencies for their nitrate-ion-based CIMS source were constant throughout the entire sampling period?
Response: Thanks for the comment.

And we have calibrated this instrument for this observations period:

First, we calibrated the instrument for this observation according to the method proposed by Kuerten et al. (2012) during the observation period. A $H_2SO_4$-based calibration factor C, with a value of $4.2\times10^9$ molecules cm-3, was obtained from a calibration using $H_2SO_4$ proceeding taking into account the diffusion loss in the sampling line.

Second, a mass dependent transmission efficiency Ti of APi-TOF was inferred in a separate experiment by depleting the reagent ions with several perfluorinated acids, following the method of Heinritzi et al. (2016).

6.   Higher resolution figures are needed.
Response: Thanks for the comment. We have updated all figures in the revised manuscript.